# Complementary congruent and opposite neurons achieve concurrent multisensory integration and segregation

Wen-Hao Zhang[1,2†], He Wang[1], Aihua Chen[3], Yong Gu[4], Tai Sing Lee[2], KY Michael Wong[1]*, Si Wu[5]*

[1]Department of Physics, Hong Kong University of Science and Technology, Hong Kong; [2]Center of the Neural Basis of Cognition, Carnegie Mellon University, Pittsburgh, United States; [3]Key Laboratory of Brain Functional Genomics, Primate Research Center, East China Normal University, Shanghai, China; [4]Institute of Neuroscience, Chinese Academy of Sciences, Shanghai, China; [5]School of Electronics Engineering and Computer Science, IDG/McGovern Institute for Brain Research, Peking-Tsinghua Center for Life Sciences, Peking University, Beijing, China

*For correspondence:
phkywong@ust.hk (KYMW);
siwu@pku.edu.cn (SW)

Present address: †Department of Mathematics, University of Pittsburgh, Pittsburgh, United States

Competing interests: The authors declare that no competing interests exist.

**Abstract** Our brain perceives the world by exploiting multisensory cues to extract information about various aspects of external stimuli. The sensory cues from the same stimulus should be integrated to improve perception, and otherwise segregated to distinguish different stimuli. In reality, however, the brain faces the challenge of recognizing stimuli without knowing in advance the sources of sensory cues. To address this challenge, we propose that the brain conducts integration and segregation concurrently with complementary neurons. Studying the inference of heading-direction via visual and vestibular cues, we develop a network model with two reciprocally connected modules modeling interacting visual-vestibular areas. In each module, there are two groups of neurons whose tunings under each sensory cue are either congruent or opposite. We show that congruent neurons implement integration, while opposite neurons compute cue disparity information for segregation, and the interplay between two groups of neurons achieves efficient multisensory information processing.
DOI: https://doi.org/10.7554/eLife.43753.001

## Introduction

To survive as an animal is to face the daily challenge of perceiving and responding fast to a constantly changing world. The brain carries out this task by gathering as much as possible information about external environments via adopting multiple sensory modalities including vision, audition, olfaction, tactile, vestibular perception, etc. These sensory modalities provide different types of information about various aspects of the external world and serve as complementary cues to improve perception in ambiguous conditions. For instance, while walking, both the visual input (optic flow) and the vestibular signal (body movement) convey useful information about heading-direction, and when integrated together, they give a more reliable estimate of heading-direction than either of the sensory modalities could deliver on its own. Indeed, experimental data has shown that the brain does integrate visual and vestibular cues to infer heading-direction and furthermore, the brain does it in an optimal way as predicted by Bayesian inference (*Fetsch et al., 2013*). Over the past years, experimental and theoretical studies verified that optimal information integration were found among many sensory modalities, for example, integration of visual and auditory cues for inferring object location (*Alais and Burr, 2004*), motion and texture cues for depth perception (*Jacobs, 1999*), visual

and proprioceptive cues for hand position (*van Beers et al., 1999*), and visual and haptic cues for object height (*Ernst and Banks, 2002*).

However, multisensory integration is only a part of multisensory information processing. While it is appropriate to integrate sensory cues from the same stimulus of interest (*Figure 1A* left), sensory cues from different stimuli need to be segregated rather than integrated in order to distinguish and recognize individual stimuli (*Figure 1A* right). In reality, the brain does not know in advance whether the cues are from the same or different objects. To accomplish the recognition task, we argue that the brain should carry out multisensory integration and segregation concurrently: a group of neurons integrates sensory cues, while the other computes the disparity information between sensory cues. The interplay between the two groups of neurons determines the final choice of integration versus segregation.

An accompanying consequence of multisensory integration is, however, that it inevitably incurs information loss of individual cues (*Figure 1*, also see Materials and methods). Consider the example of integrating the visual and vestibular cues to infer heading-direction, and suppose that both cues have equal reliability. Given that one cue yields an estimate of $\theta$ degree and the other an estimate of $-\theta$ degree, the integrated result is always 0 degree, irrespective to the value of $\theta$ (*Figure 1B*). Once the cues are integrated, the information associated with each individual cue (the value of $\theta$) is lost, and the amount of lost information increases with the extent of integration. Thus, if only multisensory integration is performed, the brain faces a chicken and egg dilemma in stimulus perception: without integrating cues, it may be unable to recognize stimuli reliably in an ambiguous environment; but once cues are integrated, the information from individual cues is lost. Concurrent multisensory integration and segregation is able to disentangle this dilemma. The information of individual cues can be recovered by using the preserved disparity information if necessary, instead of re-gathering new inputs from the external world. While there are other brain regions processing unisensory information, concurrent multisensory integration and segregation provides a unified way to achieve: (1) improved stimulus perception if the cues come from the same stimulus of interest; (2) differentiate and recognize stimuli based on individual cues with little time delay if the cues come from different stimuli of interest. This processing scheme is consistent with an experimental finding which showed that the brain can still sense the difference between cues in multisensory integration (*Wallace et al., 2004*; *Girshick and Banks, 2009*).

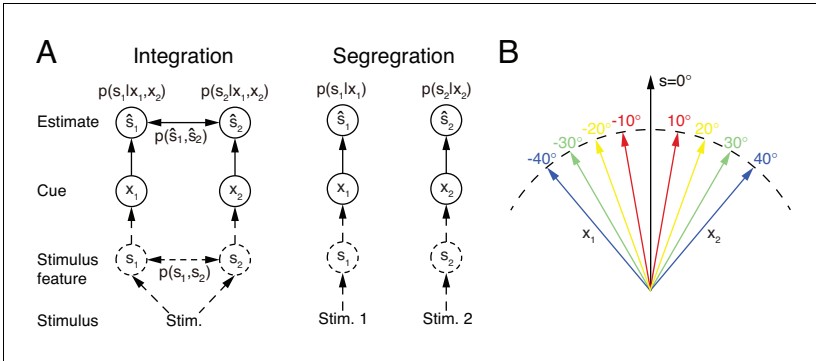

**Figure 1.** Multisensory integration and segregation. (**A**) Multisensory integration versus segregation. Two underlying stimulus features $s_1$ and $s_2$ independently generate two noisy cues $x_1$ and $x_2$, respectively. If the two cues are from the same stimulus, they should be integrated, and in the Bayesian framework, the stimulus estimation is obtained by computing the posterior $p(s_1|x_1, x_2)$ (or $p(s_2|x_1, x_2)$) utilizing the prior knowledge $p(s_1, s_2)$ (left). If two cues are from different stimuli, they should be segregated, and the stimulus estimation is obtained by computing the posterior $p(s_1|x_1)$ (or $p(s_2|x_2)$) using the single cues (right). (**B**) Information of single cues is lost after integration. The same integrated result $\hat{s} = 0°$ is obtained after integrating two cues of opposite values ($\theta$ and $-\theta$) with equal reliability. Therefore, from the integrated result, the values of single cues are unknown.
DOI: https://doi.org/10.7554/eLife.43753.002

The following figure supplement is available for figure 1:

**Figure supplement 1.** Cue disparity information is lost after integration.
DOI: https://doi.org/10.7554/eLife.43753.003

What are the neural substrates for implementing concurrent multisensory integration and segregation? Previous studies investigating the integration of visual and vestibular cues to infer heading-direction found that in each of two brain areas, namely, the dorsal medial superior temporal area (MSTd) and the ventral intraparietal area (VIP), there are two types of neurons with comparable number displaying different multisensory behaviors: congruent and opposite cells (*Figure 2*) (*Gu et al., 2008*; *Chen et al., 2013*). The tuning curves of a congruent cell in response to visual and vestibular cues are similar (*Figure 2A*), whereas the tuning curve of an opposite cell in response to a visual cue is shifted by 180 degrees (half of the period) compared to that in response to a vestibular cue (*Figure 2B*). Data analysis and modeling studies suggested that congruent neurons are responsible for cue integration (*Gu et al., 2008*; *Gu et al., 2012*; *Zhang et al., 2016*; *Ma et al., 2006*). However, the computational role of opposite neurons remains largely unknown. They do not integrate cues as their responses hardly change when a single cue is replaced by two cues with similar directions. Interestingly, however, their responses vary significantly when the disparity between visual and vestibular cues is enlarged (*Morgan et al., 2008*), indicating that opposite neurons are associated with the disparity information between cues.

In the present study, we explore whether opposite neurons are responsible for cue segregation in multisensory information processing. Experimental findings showed that many, rather than a single, brain areas exhibit multisensory processing behaviors and that these areas are intensively and reciprocally connected with each other (*Gu et al., 2008*; *Chen et al., 2013*; *Gu et al., 2016*; *Boussaoud et al., 1990*; *Baizer et al., 1991*). The architecture of these multisensory areas is consistent with the structure of a decentralized model (*Zhang et al., 2016*), where information integration naturally emerges through the interactions between distributed network modules and is robust to local failure (*Gu et al., 2012*). The decentralized model successfully reproduces almost all known phenomena observed in the multisensory integration experiments (*Fetsch et al., 2013*; *Stein and Stanford, 2008*). Thus, we consider a decentralized multisensory processing model (*Zhang et al., 2016*) in which each local processor receives a direct cue through feedforward inputs from the connected sensory modality and meanwhile, accesses information of other indirect cues via reciprocal connections between processors.

As a working example, we focus on studying the inference of heading-direction based on visual and vestibular cues. The network model consists of interconnected MSTd and VIP modules, where congruent and opposite neurons are widely found (*Gu et al., 2008*; *Chen et al., 2013*). Specifically, we propose that congruent neurons in the two brain areas are reciprocally connected with each other in the congruent manner: the closer between the preferred directions over the feedforward cue of a pair of neurons in their respective brain areas, the stronger their connection is, and this connection profile encodes effectively the prior knowledge about the two cues coming from the same stimulus. On the other hand, opposite neurons in the two brain areas are reciprocally connected in the opposite manner: the further away between the preferred directions over the feedforward

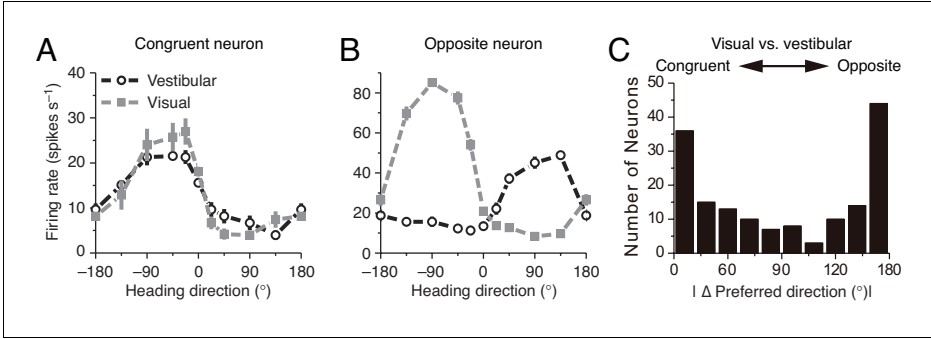

**Figure 2.** Congruent and opposite neurons in MSTd. Similar results were found in VIP (*Chen et al., 2011*). (A–B) Tuning curves of a congruent neuron (A) and an opposite neuron (B). The preferred visual and vestibular directions are similar in (A) but are nearly opposite by 180˚ in (B). (C) The histogram of neurons according to their difference between preferred visual and vestibular directions. Congruent and opposite neurons are comparable in numbers. (A–B) are adapted from *Gu et al. (2008)*, (C) from *Gu et al. (2006)*.
DOI: https://doi.org/10.7554/eLife.43753.004

cue of a pair of neurons in their respective brain areas (the maximal difference is 180 degree), the stronger their connection is. Our model reproduces the tuning properties of opposite neurons, and verifies that opposite neurons encode the disparity information between cues. Furthermore, we demonstrate that this disparity information, in coordination with the integration result of congruent neurons, enables the neural system to assess the validity of cue integration and to recover the lost information of individual cues if necessary. Our study sheds light on our understanding of how the brain achieves multisensory information processing efficiently.

## Results

### Probabilistic models of multisensory processing

The brain infers stimulus information based on ambiguous sensory cues. We therefore formulate the multisensory processing problem in the framework of probabilistic inference, and as a working example, we focus on studying the inference of heading-direction based on visual and vestibular cues.

#### Probabilistic model of multisensory integration

To begin with, we introduce the probabilistic model of multisensory integration. Suppose two stimulus features $\{s_m\}$ generate two sensory cues $\{x_m\}$, for $m = 1, 2$ (the visual and vestibular cues), respectively (*Figure 1A*), and we denote the corresponding likelihood functions as $p(x_m|s_m)$. The task of multisensory processing is to infer $\{s_m\}$ based on $\{x_m\}$. $x_m$ is referred to as the direct cue of $s_m$ (e.g. the visual cue to MSTd) and $x_l$ $(l \neq m)$ the indirect cue of $s_m$ (e.g. the vestibular cue to MSTd).

Since heading-direction is a circular variable in the range of $(-\pi, \pi]$, we adopt the von Mises, rather than the Gaussian, distribution to carry out the theoretical analysis. In the form of the von Mises distribution, the likelihood function is given by

$$
\begin{aligned}
p(x_m|s_m) &= [2\pi I_0(\kappa_m)]^{-1}\exp[\kappa_m\cos(x_m - s_m)]\\
&\equiv \mathcal{M}(x_m; s_m, \kappa_m),
\end{aligned}
\tag{1}
$$

where $I_0(\kappa)$ is the modified Bessel function of the first kind and order zero, and acts as the normalization factor. $s_m$ is the mean of the von Mises distribution, that is the mean value of $x_m$. $\kappa_m$ is a positive number characterizing the concentration of the distribution, and controls the reliability of cue $x_m$.

The prior $p(s_1, s_2)$ describes the probability of concurrence of stimulus features $(s_1, s_2)$. In the literature, the study of integration and segregation was often formulated as the issue of causal inference (*Sato et al., 2007*; *Körding et al., 2007*; *Shams and Beierholm, 2010*). In general, the prior of causal inference consists of more than one components, each corresponding to the causal structure describing the relation between the multiple stimuli. In this study, we consider a single-component integration prior which has been used in several multisensory integration studies (*Bresciani et al., 2006*; *Roach et al., 2006*; *Sato et al., 2007*; *Zhang et al., 2016*), and it is sufficient to demonstrate the role played by the congruent and opposite neurons, yet retaining a simpler mathematical framework (see more discussions in Conclusions and Discussions). The integration prior is

$$
\begin{aligned}
p(s_1, s_2) &= (2\pi)^{-1}\mathcal{M}(s_1 - s_2; 0, \kappa_s)\\
&= \left[(2\pi)^2 I_0(\kappa_s)\right]^{-1}\exp[\kappa_s\cos(s_1 - s_2)].
\end{aligned}
\tag{2}
$$

This prior reflects that the two stimulus features from the same stimulus tend to have similar values. The parameter $\kappa_s$ specifies the concurrence probability of two stimulus features, and determines the extent to which the two cues should be integrated. In the limit $\kappa_s \to \infty$, it will lead to full integration (see, e.g. *Ernst and Banks, 2002*). Note that the marginal prior $p(s_m)$ is a uniform distribution according to the definition.

It has been revealed that in the congruent cueing condition, the brain integrates visual and vestibular cues to infer heading-direction in a manner close to Bayesian inference (*Gu et al., 2008*; *Chen et al., 2013*). Following Bayes' theorem, optimal multisensory integration is achieved by computing the posterior of two stimuli according to

$$
p(s_1, s_2|x_1, x_2) \propto p(x_1|s_1)p(x_2|s_2)p(s_1, s_2).
$$

Since the calculations of the two stimuli are exchangeable, hereafter we only present the results for $s_1$. The posterior of $s_1$ is calculated through marginalizing the joint posterior in the above equation,

$$
\begin{aligned}
p(s_1|x_1,x_2) \;&\propto p(x_1|s_1)\int_{-\pi}^{\pi} p(x_2|s_2)p(s_1,s_2)ds_2 \\
&\propto p(s_1|x_1)p(s_1|x_2) \\
&\approx \mathcal{M}(s_1;x_1,\kappa_1)\mathcal{M}(s_1;x_2,\kappa_{2s}),
\end{aligned}
\tag{3}
$$

where we have used the conditions that the marginal prior distributions of $s_m$ and $x_m$ are uniform, that is $p(s_m)=p(x_m)=(2\pi)^{-1}$. Note that $p(s_1|x_2)\propto \int p(x_2|s_2)p(s_1,s_2)ds_2$ is approximated to be $\mathcal{M}(s_1;x_2,\kappa_{2s})$ through equating the mean resultant length of distribution (*Equation 13*) (*Mardia and Jupp, 2009*).

The above equation indicates that in multisensory integration, the posterior of a stimulus given combined cues is equal to the product of the posteriors given the individual cues. Notably, although $x_1$ and $x_2$ are generated independently by $s_1$ and $s_2$ (since the visual and vestibular signal pathways are separated), $x_2$ also provides information of $s_1$ due to the correlation between $s_1$ and $s_2$ specified in the prior.

Finally, since the product of two von Mises distributions is again a von Mises distribution, the posterior distribution is $p(s_1|x_1,x_2)=\mathcal{M}(s_1;\hat{s}_1,\hat{\kappa}_1)$, whose mean and concentration can be obtained from its moments given by

$$
\hat{\kappa}_1 e^{j\hat{s}_1} = \kappa_1 e^{jx_1} + \kappa_{2s} e^{jx_2},
\tag{4}
$$

where $j$ is an imaginary number. *Equation 4* is the result of Bayesian optimal integration in the form of von Mises distributions, and they are the criteria to judge whether optimal cue integration is achieved in the neural system. A link between the Bayesian criteria for von Mises and Gaussian distributions is presented in Appendix 2.

*Equation 4* indicates that the von Mises distribution of a circular variable can be interpreted as a vector in a two-dimensional space with its mean and concentration representing the angle and length of the vector, respectively (*Figure 3A*). In this interpretation, the product of two von Mises distributions can be represented by the summation of the corresponding two vectors. Thus, optimal multisensory integration is equivalent to vector summation (see *Equation 4*), with each vector representing the posterior of the stimulus given each cue (the sum of the two green vectors yields the blue vector in *Figure 3B*).

## Probabilistic model of multisensory segregation

The above probabilistic model for multisensory integration assumes that sensory cues are originated from the same stimulus. In case they come from different stimuli, the cues need to be segregated, and the neural system needs to infer stimuli based on individual cues. In practice, the brain needs to differentiate these two situations. In order to achieve reliable multisensory processing, we propose that while integrating sensory cues, the neural system simultaneously extracts the disparity information between cues, so that with this complementary information, the neural system can assess the validity of cue integration.

An accompanying consequence of multisensory integration is that the stimulus information associated with individual cues is lost once they are integrated (see *Figure 1—figure supplement 1*). Hence besides assessing the validity of integration, extracting both congruent and disparity information by simultaneous integration and segregation enables the system to recover the lost information of individual cues if needed.

The disparity information of stimulus one obtained from the two cues is defined to be

$$
p_d(s_1|x_1,x_2) \propto p(s_1|x_1)/p(s_1|x_2),
\tag{5}
$$

which is the ratio between the posterior given two cues and hence measures the discrepancy between the estimates from different cues. By taking the expectation of $\log p_d$ over the distribution $p(s_1|x_1)$, it gives rise to the Kullback-Leibler divergence between the two posteriors given each cue.

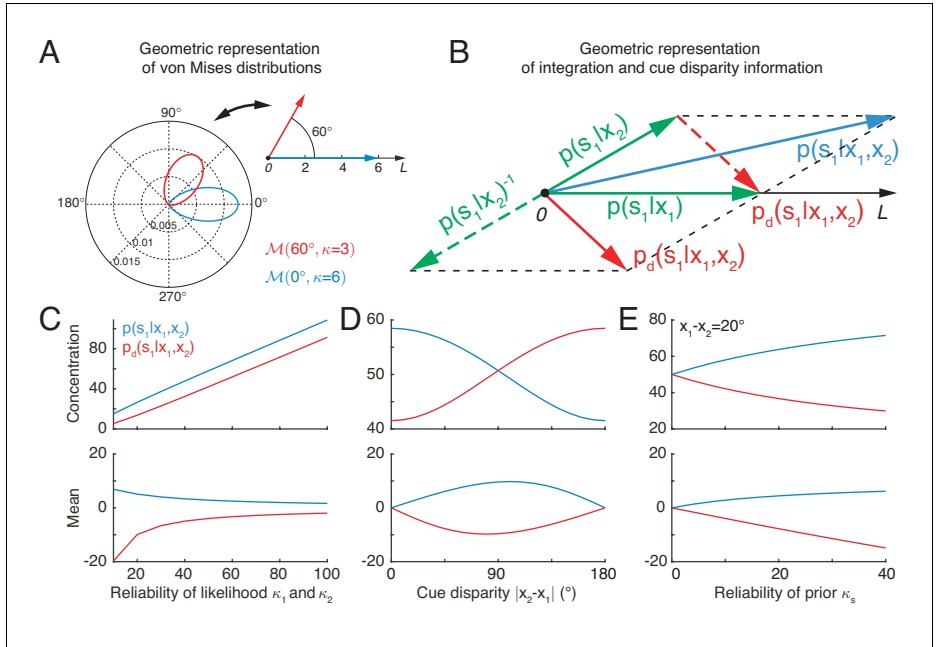

**Figure 3.** Geometric interpretation of multisensory processing of circular variables. (A) Two von Mises distributions plotted in the polar coordinate (bottom-left) and their corresponding geometric representations (top-right). A von Mises distribution can be represented as a vector, with its mean and concentration corresponding to the angle and length of the vector, respectively. (B) Geometric interpretation of cue integration and the cue disparity information. The posteriors of $s_1$ given single cues are represented by two vectors (green). Cue integration (blue) is the sum of the two vectors (green), and the cue disparity information (red) is the difference of the two vectors. (C–E) The mean and concentration of the integration (blue) and the cue disparity information (red) as a function of the cue reliability (C), cue disparity (D), and reliability of prior (E). In all plots, $\kappa_s = 50$, $\kappa_1 = \kappa_2 = 50$, $x_1 = 0°$ and $x_2 = 20°$, except that the variables are $\kappa_1 = \kappa_2$ in C, $x_2$ in D, and $\kappa_s$ in E.
DOI: https://doi.org/10.7554/eLife.43753.005

This disparity measure was also used to discriminate alternative moving directions in *Jazayeri et al. (2006)*.

Utilizing the property of the von Mises distribution and the periodicity of heading directions ($-\cos(s_1 - x_2) = \cos(s_1 - x_2 - \pi)$), *Equation 5* can be re-written as

$$\begin{aligned} p_d(s_1|x_1,x_2) &\propto p(s_1|x_1)p(s_1|x_2 + \pi) \\ &\propto \mathcal{M}(s_1;x_1,\kappa_1)\mathcal{M}(s_1;x_2 + \pi,\kappa_{2s}). \end{aligned} \tag{6}$$

Thus, the disparity information between two cues can also be expressed as the product of the posterior given the direct cue and the posterior given the indirect cue with the cue direction shifted by $\pi$. Indeed, analogous to the derivation of *Equation 3*, *Equation 6* can be deduced in the same framework as multisensory integration but with the stimulus prior $p(s_1, s_2)$ being modified by a shift $\pi$ in the angular difference. Similarly, $p_d(s_1|x_1,x_2) = \mathcal{M}(s_1;\Delta\hat{s}_1,\Delta\hat{\kappa}_1)$ whose mean and concentration can be derived as

$$\Delta\hat{\kappa}_1 e^{j\Delta\hat{s}_1} = \kappa_1 e^{jx_1} - \kappa_{2s} e^{jx_2}. \tag{7}$$

The above equation is the criteria to judge whether the disparity information between two cues is encoded in the neural system.

Similar to the geometrical interpretation of multisensory integration, multisensory segregation is interpreted as vector subtraction (the subtraction between two green vectors yields the red vector in *Figure 3B*). This enables us to assess the validity of multisensory integration. When the two vectors representing the posteriors given the individual cues have small disparity, that is the estimates from individual cues tend to support each other, the length of the summed vector is long, implying

that the posterior of cue integration has a strong confidence, whereas the length of the subtracted vector is short, implying that the weak confidence of two cues are disparate (*Figure 3D*). If the two vectors associated with the individual cues have a large disparity, the interpretation becomes the opposite (*Figure 3D*). Thus, by comparing the lengths of the summed and subtracted vectors, the neural system can assess whether two cues should be integrated or segregated.

*Figure 3C and E* further describes the integration and segregation behaviors when the model parameters vary. As shown in *Figure 3C*, when the likelihoods have weak reliabilities, the network estimate relies more on the prior. Since the prior encourages integration of the two stimuli, the posterior estimate of stimulus one becomes more biased towards cue 2. At the same time, the mean of the disparity information is biased toward the angular difference of the likelihood peaks. On the other hand, when the likelihoods are strong, the network estimate relies more on the likelihood, and the posterior estimate of stimulus one becomes less biased toward cue 2. The behavior when the prior concentration $\kappa_s$ varies can be explained analogously (*Figure 3E*).

A notable difference between von Mises distribution and Gaussian distribution is that the concentration of integration and disparity information changes with cue disparity in von Mises distribution (*Figure 3D*), while they are fixed in Gaussian distribution (*Ernst, 2006*).

## Neural implementation of cue integration and segregation

Before introducing the neural circuit model, we first describe intuitively how opposite neurons encode the cue disparity information and the motivation of the proposed network structure.

Optimal multisensory integration computes the posterior of a stimulus given combined cues according to *Equation 3*, which is equivalent to solving the equation $\ln p(s_1|x_1, x_2) = \ln p(s_1|x_1) + \ln p(s_1|x_2)$. Ma *et al.* found that under the conditions that neurons fire independent Poisson spikes, the optimal integration can be achieved by combining the neuronal responses under single cue conditions, that is $\mathbf{r}_j(x_1, x_2) = \mathbf{r}_j(x_1) + \mathbf{r}_j(x_2)$ (see details in Materials and methods), where $\mathbf{r}(x_1, x_2)$ and $\mathbf{r}(x_m)$ are the responses of a population of neurons to the combined and single cues respectively (*Ma et al., 2006*). Ma *et al.* further demonstrated that such a response property can be approximately achieved in a biological neural network. Similarly, multisensory segregation computes the disparity information between cues according to $\ln p_d(s_1|x_1, x_2) = \ln p(s_1|x_1) + \ln p(s_1|x_2 + \pi)$ (see *Equation 6*). Analogous to multisensory integration, multisensory segregation can be achieved by $\mathbf{r}_j(x_1, x_2) = \mathbf{r}_j(x_1) + \mathbf{r}_{j'}(x_2)$, where the preferred stimulus of neurons satisfying $\theta_{j'} = \theta_j + \pi$ (see details in Materials and methods). That is, the neurons combine the responses to the direct cue and the responses to the indirect cue but shifted to opposite direction. This inspires us to consider a network model where the inputs of indirect cue received by opposite neurons are shifted to opposite direction via connections. Below, we present the network model and demonstrate that the opposite neurons emerge from the connectivity and are able to achieve cue segregation.

### The decentralized neural network model

The neural circuit model we consider has the decentralized structure (*Zhang et al., 2016*), in the sense that it consists of two reciprocally connected modules (local processors), representing MSTd and VIP respectively (*Figure 4A*). Each module carries out multisensory processing via cross-talks between modules. This decentralized architecture achieves integration in a distributed way and is robust to local failure, and it agrees with the experimental findings that neurons in MSTd and VIP both exhibit multisensory responses and that the two areas are abundantly connected with each other (*Boussaoud et al., 1990*; *Baizer et al., 1991*). Below we only describe the key features of the decentralized network model, and its detailed mathematical description is presented in Materials and methods (*Equations 16-22*).

At each module, there exist two groups of excitatory neurons: congruent and opposite neurons (blue and red circles in *Figure 4A* respectively), and they have the same number of neurons, as supported by experiments (*Figure 2C*) (*Chen et al., 2011*; *Gu et al., 2006*). Each group of neurons is modeled as a continuous attractor neural network (CANN), mimicking the encoding of heading-direction in neural systems (*Zhang, 1996*; *Wu et al., 2008*). In CANN, each neuron is uniquely identified by its preferred heading direction $\theta$ with respect to the direct cue conveyed by feedforward inputs. The neurons in the same group are recurrently connected, and the recurrent connection

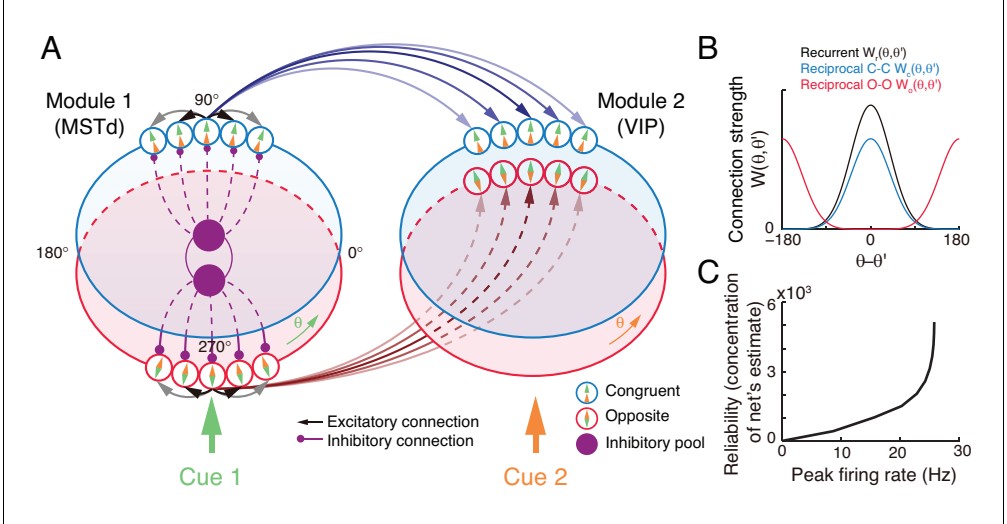

**Figure 4.** The decentralized neural circuit model for multisensory processing. (**A**) The network consists of two modules, which can be regarded as MSTd and VIP respectively. Each module has two groups of excitatory neurons, congruent (blue circles) and opposite neurons (red circles). Each group of excitatory neurons are connected recurrently with each other, and they are all connected to an inhibitory neuron pool (purple disk) to form a continuous attractor neural network. Each module receives a direct cue through feedforward inputs. Between modules, congruent neurons are connected in the congruent manner (blue arrows), while opposite neurons are connected in the opposite manner (brown lines). (**B**) Connection profiles between neurons. Black line is the recurrent connection pattern between neurons of the same type in the same module. Blue and red lines are the reciprocal connection patterns between congruent and opposite neurons across modules respectively. (**C**) The reliability of the network's estimate of a stimulus is encoded in the peak firing rate of the neuronal population. Typical parameters of network model: $\omega = 3 \times 10^{-4}$, $J_{int} = 0.5$, $J_{rc} = 0.3J_c$, $J_{rp} = 0.5J_{rc}$, $I_b$ and $F$ in **Equation 22** are 1 and 0.5 respectively.

DOI: https://doi.org/10.7554/eLife.43753.006

strength between neurons $\theta$ and $\theta'$ is modeled as a von Mises function decaying with the disparity between two neurons' preferred directions $|\theta - \theta'|$ (**Figure 4B** black line and **Equation 17**). In the model, the recurrent connection strength is not very strong to support persistent activities after switching off external stimuli, because no persistent activity is observed in multisensory areas. Moreover, neuronal responses in the same group are normalized by the total activity of the population (**Equation 20**), called divisive normalization (**Carandini and Heeger, 2012**), mimicking the effect of a pool of inhibitory neurons (purple disks in **Figure 4B**). Each group of neurons has its individual inhibitory neuron pool, and the two pools of inhibitory neurons in the same module share their overall activities (**Equation 21**), which intends to introduce mutual inhibition between congruent and opposite neurons.

Between modules, neurons of the same type are reciprocally connected with each other (**Figure 4A–B**). For congruent neurons, they are connected with each other in the congruent manner (**Equation 18** and **Figure 4B** blue line), that is, the more similar their preferred directions are, the stronger the neuronal connection is. For opposite neurons, they are connected in the opposite manner (**Equation 19** and **Figure 4B** red line), that is, the more different their preferred directions are, the stronger the neuronal connection is. Since the maximum difference between two circular variables is $\pi$, an opposite neuron in one module preferring $\theta$ has the strongest connection to the opposite neuron preferring $\theta + \pi$ in the other module. This agrees with our intuitive understanding as described above (as suggested by **Equation 6**): to calculate the disparity information between two cues, the neuronal response to the combined cues should integrate its responses to the direct cue and its response to the indirect one but with the cue direction shifted by $\pi$ (through the offset reciprocal connections). We set the connection profile between the opposite neurons to be of the same strength and width as that between the congruent ones (comparing **Equations 18 and 19**), ensuring

that the tuning functions of the opposite neurons have the similar shape as those of the congruent ones, as observed in the experimental data (*Chen et al., 2011*).

When sensory cues are applied, the neurons combine the feedforward, recurrent, and reciprocal inputs to update their activities (*Equation 16*), and the multisensory integration and segregation will be accomplished by the reciprocal connections between network modules. The results are presented below.

## Tuning properties of congruent and opposite neurons

Simulating the neural circuit model, we first checked the tuning properties of neurons. The simulation results for an example congruent neuron and an example opposite neuron in module 1 responding to single cues are presented in *Figure 5*. It shows that the congruent neuron, in response to either cue 1 or cue 2, prefers the same direction ($-90°$) (*Figure 5A*), whereas the opposite neuron, while preferring $-90°$ for cue 1, prefers $90°$ for cue 2 (*Figure 5B*). Thus, the tuning properties of congruent and opposite neurons naturally emerge through the network dynamics.

We further checked the responses of neurons to combined cues and found that when there is no disparity between the two cues, the response of a congruent neuron is enhanced compared to the single cue conditions (green line in *Figure 5A*), whereas the response of an opposite neuron is suppressed compared to its response to the direct cue (green line in *Figure 5B*). These properties agree with the experimental data (*Gu et al., 2008*; *Chen et al., 2013*) and is also consistent with the interpretation that the integrated and segregated amplitudes are respectively proportional to the vector sum and difference in *Figure 3*. Following the experimental protocol (*Morgan et al., 2008*), we also plotted the bimodal tuning curves of the example neurons in response to the combined cues of varying reliability, and observed that when cue 1 has a relatively high reliability, the bimodal responses of both neurons are dominated by cue 1 (*Figure 5C–D*), indicating that the neuronal firing rates are affected more significantly by varying the angle of cue 1 than by that of cue 2, whereas when the reliability of cue 1 is reduced, the result becomes the opposite (*Figure 5E–F*). These behaviors agree with the experimental observations (*Morgan et al., 2008*).

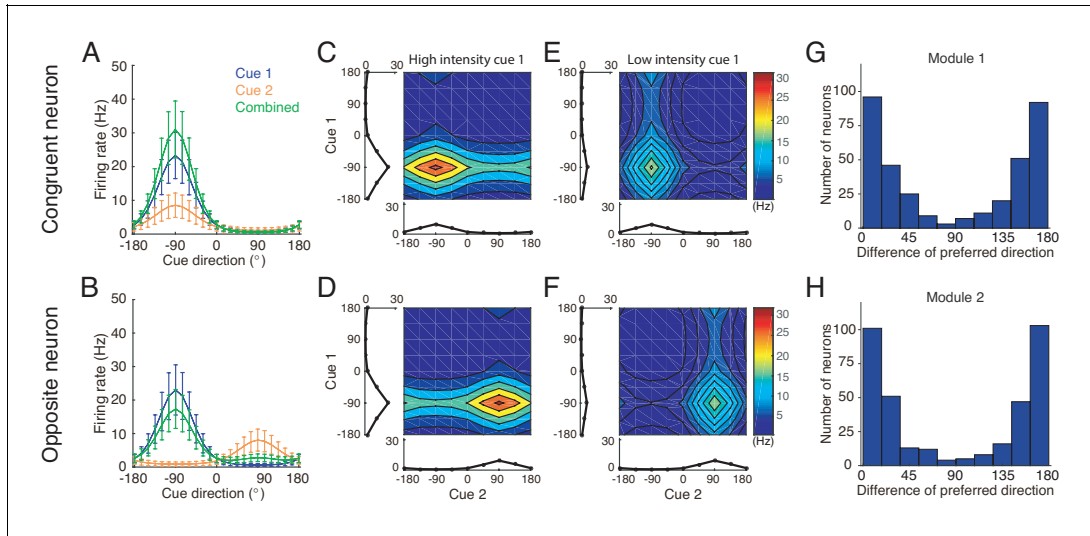

**Figure 5.** Tuning properties of congruent and opposite neurons in the network model. (A–B) The tuning curves of an example congruent neuron (A) and an example opposite neuron (B) in module 1 under three cueing conditions. (C–D) The bimodal tuning properties of the example congruent (C) and the example opposite (D) neurons when cue 1 has relatively higher reliability than cue 2 in driving neurons in module 1, with $\alpha_1 = 0.58\alpha_2$, where $\alpha_m$ is the amplitude of cue $m$ given by *Equation 22*. The two marginal curves around each contour plot are the unimodal tuning curves. (E–F) Same as (C–D), but cue 1 has a reduced reliability with $\alpha_1 = 0.12\alpha_2$. (G–H) The histogram of the differences of neuronal preferred directions with respect to two cues in module 1 (G) and module 2 (H), when the reciprocal connections across network modules contain random components of roughly the same order as the connections. Parameters: (A–B) $\alpha_1 = 0.35U_0$, and $\alpha_2 = 0.8U_0$; (C–F) $\alpha_2 = 1.5U_0$ in (C–D) while $\alpha_1 = 0.1U_0$ in (E–F). Other parameters are the same as those in *Figure 4*.

DOI: https://doi.org/10.7554/eLife.43753.007

Apart from the congruent and opposite neurons, the experiments also found that there exist a portion of neurons, called intermediate neurons, whose preferred directions to different cues are neither exactly the same nor the opposite, but rather have differences in between 0° and 180° (*Gu et al., 2006*; *Chen et al., 2011*). We found that by considering the realistic imperfectness of neuronal reciprocal connections (e.g. adding random components in the reciprocal connections in *Equations (18 and 19)*, see Materials and methods), our model reproduced the distribution of intermediate neurons as observed in the experiment (*Figure 5G–H*) (*Gu et al., 2006*; *Chen et al., 2011*).

## Cue integration and segregation via congruent and opposite neurons

In response to the noisy inputs in a cueing condition, the population activity of the same group of neurons in a module exhibits a bump-shape (*Figure 6A*), and the position of the bump is interpreted as the network's estimate of the stimulus (*Figure 6B*) (*Deneve et al., 1999*; *Wu et al., 2002*; *Wu et al., 2008*). In a single instance, we used the population vector to read out the stimulus value

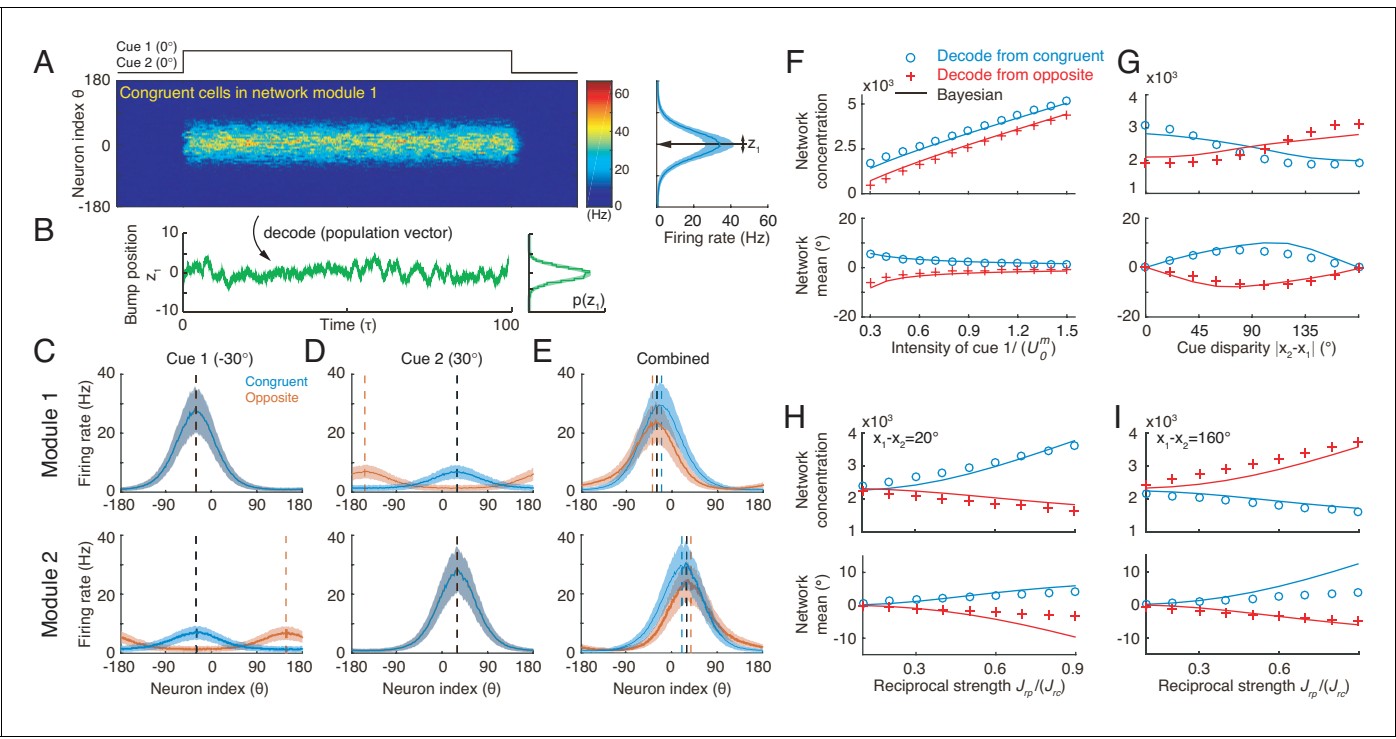

**Figure 6.** Optimal cue integration and segregation collectively emerge in the neural population activities in the network model. (**A**) Illustration of the population response of congruent neurons in module 1 when both cues are presented. Color indicates firing rate. Right panel is the temporal average firing rates of the neural population during cue presentation, with shaded region indicating the standard deviation (SD). Note that the neuron index $\theta$ refers to the preferred direction with respect to the direct cue conveyed by feedforward inputs. (**B**) The position of the population activity bump at each instance is interpreted as the network's estimate of the stimulus, referred to as $z_1$, which is decoded by using population vector. Right panel is the distribution of the decoded network's estimate during cue presentation. (**C–E**) The temporal average population activities of congruent (blue) and opposite (red) neurons in module 1 (top row) and module 2 (bottom row) under three cueing conditions: only cue 1 is presented (**C**), only cue 2 is presented (**D**), and both cues are simultaneously presented (**E**). (**F–I**) Comparing the estimates from congruent and opposite neurons in module 1 with the theoretical predictions, with varying cue intensity (**F**), with varying cue disparity (**G**), and with varying reciprocal connection strength between modules (**H** and **I**). Symbols: network results; lines: theoretical prediction. The theoretical predictions for the estimates of congruent and opposite neurons are obtained by *Equations 4* and *7*. Parameters: (**A–E**) $\alpha_1 = \alpha_2 = 0.35U_0$; (**F**) $\alpha_2 = 0.7U_0$; (**G–I**) $\alpha_1 = \alpha_2 = 0.7U_0$, and others are the same as those in *Figure 4*. In (**F–H**), $x_1 = 0°$, $x_2 = 20°$ and in (**I**), $x_1 = 0°$, $x_2 = 160°$.

DOI: https://doi.org/10.7554/eLife.43753.008

The following figure supplements are available for figure 6:

**Figure supplement 1.** Illustration of decoded joint distributions from congruent and opposite neurons.
DOI: https://doi.org/10.7554/eLife.43753.009

**Figure supplement 2.** Test of network's performance.
DOI: https://doi.org/10.7554/eLife.43753.010

(*Equation 23*) (*Georgopoulos et al., 1986*). The statistics of the bump position sampled from a collection of instances reflects the posterior distribution of the stimulus estimated by the neural population under the given cueing condition.

To validate the hypothesis that congruent and opposite neurons are responsible for cue integration and segregation respectively, we carried out simulations following the protocol in multisensory experiments (*Fetsch et al., 2013*), that is, we first applied individual cues to the network and decoded the network's estimate of the stimulus through population vector (see details in Materials and methods). With these results, the theoretical predictions for cue integration and segregation were calculated according to *Equations 4* and *7,* respectively; we then applied the combined cues to the network, decoded the network's estimate, and compared them with the theoretical predictions.

Let us first look at the network's estimate under single cue conditions. Consider the case that only cue 1 is presented to module 1 at −30˚. The population activities of congruent and opposite neurons at module 1 are similar, both centered at −30˚ (*Figure 6C* top), since both types of neurons receive the same feedforward input. On the other hand, in module 2, congruent neurons' responses are centered at −30˚, while opposite neurons' responses are centered at 150˚ due to the offset reciprocal connections (*Figure 6C* bottom). Similar population activities exist under cue 2 condition (*Figure 6D*).

We further look at the the network's estimate under the combined cue condition. Consider the case that cues 1 and 2 are simultaneously presented to the network at the directions −30˚ and 30˚ respectively. Then the disparity between the two cues is 60˚, which is less than 90˚. Compared with single cue conditions, the responses of congruent neurons are enhanced (comparing *Figure 6E* with *Figure 6C-D*), reflecting the increased reliability of the estimate after cue integration. Indeed, the decoded distribution from congruent neurons sharpens in the combined cue condition and moves to a location between cue 1 and cue 2 (*Figure 6—figure supplement 1* green), which is a typical phenomenon associated with cue integration. In contrast, with combined cues, the responses of opposite neurons are suppressed compared with those of the direct cue (comparing *Figure 6E* with *Figure 6C-D*). Certainly, the distribution of cue disparity information decoded from opposite neurons in combined cue condition is wider than that that under the direct cue condition (*Figure 6—figure supplement 1* purple). Note that when the cue disparity is larger than 90˚, the relative response of congruent and opposite neurons will be reversed (results are not shown here).

To demonstrate that the network implements cue integration and segregation and how the network encodes the probabilistic model (*Equations 1 and 2*), we changed a parameter at a time, and then compared the decoded results from congruent and opposite neurons with the theoretical predictions. *Figure 6F–I* indicates that the network indeed implements optimal integration and segregation. Moreover, comparing the network results with the results of the probabilistic model, we could find the analogy that the input intensity encodes the reliability of the likelihood (*Equation 1*, comparing *Figure 6F* with *Figure 3C*), and the reciprocal connection strength effectively represents the reliability of the integration prior (*Equation 2*, comparing *Figure 6H* with *Figure 3E*), which is consistent with a previous study (*Zhang et al., 2016*). We further systematically changed the network and input parameters over a large parameter region and compare the network results with theoretical predictions. Our results indicated that the network model achieves cue integration and segregation robustly over a large range of parameters (*Figure 6—figure supplement 2*), as long as the connection strengths are not so large that winner-take-all happens in the network model.

## Concurrent multisensory processing

The above results elucidate that congruent neurons integrate cues, whereas opposite neurons compute the disparity between cues. Based on these complementary information, the brain can access the validity of cue integration and can also recover the stimulus information associated with single cues lost due to integration. Below, rather than exploring the detailed neural circuit models, we demonstrate that the brain has resources to implement these two operations based on the activities of congruent and opposite neurons.

## Assessing integration vs. segregation

The competition between congruent and opposite neurons can determine whether the brain should integrate or segregate two cues. *Figure 7A* displays how the mean firing rates of two types of neurons change with the cue disparity, which shows that the activity of congruent neurons decreases with the disparity, whereas the activity of opposite neurons increases with the disparity, and they are equal at the disparity value of 90°. The brain can judge the validity of integration based on the competition between these two groups of neurons (see more remarks in Conclusions and Discussions). Specifically, the group of congruent neurons wins when the cue disparity is small, indicating the choice of integration, and the group of opposite neurons wins when the cue disparity is large, indicating the choice of segregation. The decision boundary is at the disparity of 90°, if the activities of congruent and opposite neurons have equal weights in decision-making. In reality, however, the brain may assign different weights to congruent and opposite neurons and realize a decision boundary at the position satisfying the statistics of inputs (*Figure 7B*).

## Recovering the single cue information

Once the decision for cue segregation is reached, the neural system at each module needs to decode the stimulus based purely on the direct cue, and ignores the irrelevant indirect one. Through combining the complementary information from congruent and opposite neurons, the neural system can recover the stimulus estimates lost in integration, without re-gathering new inputs from lower brain areas if needed (see more remarks in Conclusions and Discussions).

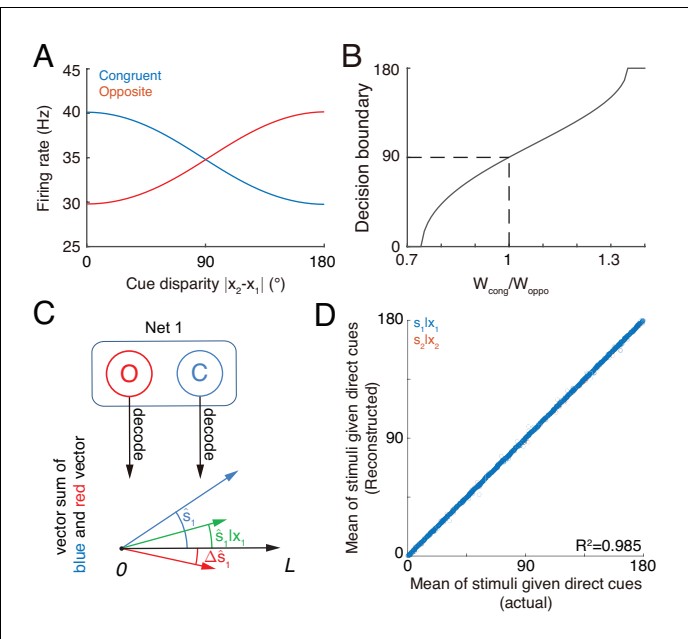

**Figure 7.** Concurrent multisensory processing with congruent and opposite neurons. (**A–B**) Accessing integration versus segregation through the joint activity of congruent and opposite neurons. (**A**) The firing rate of congruent and opposite neurons exhibit complementary changes with cue disparity $x_1 - x_2$. (**B**) The decision boundary of the competition between congruent and opposite neurons changes with read out weight from congruent $W_{cong}$ and opposite neurons $W_{oppo}$. It is given by the value of $x_1 - x_2$ at which $W_{cong}r_m^c = W_{oppo}r_m^o$. Dashed line is when $W_{cong} = W_{oppo}$, the decision boundary is at 90°. (**C–D**) Recovering single cue information from two types of neurons. (**C**) Illustration of recovering through the joint activities of congruent (blue) and opposite (red) neurons under the combined cue condition. We decoded the estimate from congruent and opposite neurons respectively, and then vector sum the decoded results recovering the single cue information. (**D**) Comparing the recovered mean of the stimulus given the direct cue with the actual value. Parameters: those in (**A–B**) are the same as those in *Figure 6A*, and those in D are the same as those in *Figure 6—figure supplement 2*.

DOI: https://doi.org/10.7554/eLife.43753.011

According to *Equations 3 and 6*, the posterior distribution of the stimulus given the direct cue can be recovered by

$$\ln p(s_1|x_1) = [\ln p(s_1|x_1,x_2) + \ln p_d(s_1|x_1,x_2)]/2. \tag{8}$$

As suggested in *Ma et al. (2006)* and *Jazayeri et al. (2006)*, the above operation can be realized by considering neurons receiving the activities of congruent neurons (representing $\ln p(s_1|x_1,x_2)$, *Figure 7C* blue) and opposite neurons (representing $\ln p_d(s_1|x_1,x_2)$, *Figure 7C* red) as inputs and generate Poisson spikes, such that the location of population responses and the summed activity encode respectively the mean and variance of the posterior $p(s_1|x_1)$ (*Figure 7C* green).

Without actually building a neural circuit model, we decoded the stimulus by utilizing the activities of congruent and opposite neurons according to *Equation 8*, and compared the recovered result with the estimate of a module when only the direct cue is presented (see the detail in Materials and methods). *Figure 7D* further shows that the recovering agrees with actual distribution and is robust against a variety of parameters ($R^2 = 0.985$). Thus, through combining the activities of congruent and opposite neurons, the neural system can recover the lost stimulus information from direct cues if necessary.

## Experimental predictions

The key structure of our network model can be tested in experiments. For instance, we may measure the correlations between congruent neurons and between opposite neurons across modules, and the correlations between congruent and opposite neurons within and across modules. According to the connection structure of our model, the averaged correlations between the same type of neurons across modules are positive due to the excitatory connections between them, whereas the averaged correlations between different types of neurons within and across modules are negative due to the competition between them. We may also inactivate one type of neurons in one module and observe the neurons in the other module, the activity of the same type of neurons is suppressed, whereas the activity of the other type of neurons is enhanced.

Furthermore, our hypothesis on the computational role of opposite neurons can be evaluated by experiments. Through recording the activities of individual congruent neurons in awake monkeys when the monkeys are performing heading-direction discrimination, previous studies demonstrated that congruent neurons implement optimal cue integration in the congruent cueing condition (*Gu et al., 2008*; *Chen et al., 2013*). We can carry out a similar experiment to check whether opposite neurons encode the cue disparity information. The task is to discriminate whether the disparity from two cues, $x_1 - x_2$, is either smaller or larger than 0°. To rule out the influence of the change of integrated direction to the activities of neurons, we fix the center of two cues, for example, the center is fixed at 0°, that is $x_1 + x_2 = 0°$, but the disparity between cues $x_1 - x_2$ varies over trials. *Figure 8A* plots the responses of an example opposite neuron and an example congruent neuron respectively in our model with respect to the cue disparity $x_1 - x_2$. It shows that the firing rate of the opposite neurons changes much more significantly with the cue disparity than that of the congruent neuron, suggesting that the opposite neuron's response might be more informative to the change of cue disparity compared with a congruent neuron. To quantify how the activity of a single neuron can be used to discriminate the cue disparity, we apply receiver-operating-characteristics (ROC) analysis to construct the neurometric function (*Figure 8B*), which measures the fraction of correct discrimination (see Materials and methods). Indeed, the opposite neurons can discriminate the cue disparity much finer than congruent neurons (*Figure 8C*). In addition, our model also reproduces the same discrimination task studied in *Gu et al. (2008)* and *Chen et al. (2013)*, that is to discriminate whether the heading-direction is on the left or right hand side of a reference direction under different cueing conditions (*Figure 8—figure supplement 1*).

## Discussion

Animals face challenges of processing information fast in order to survive in natural environments, and over millions of years of evolution, the brain has developed efficient strategies to handle these challenges. In multisensory processing, such a challenge is to integrate/segregate multisensory sensory cues rapidly without knowing in advance whether these cues are from the same or different

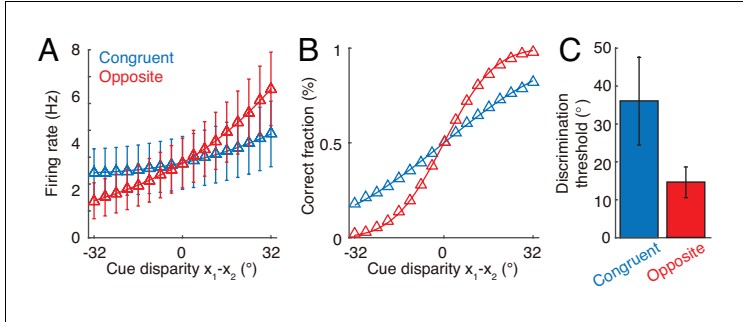

**Figure 8.** Discrimination of cue disparity by single neurons. (**A**) The tuning curve of an example congruent (blue) and opposite (red) neuron with respect to cue disparity $x_1 - x_2$. In the tuning with respect to cue disparity, the mean of two cues was always at 0°, that is $x_1 + x_2 = 0$, while their disparity $x_1 - x_2$ was varied from $-32$° to 32° with a step of 4°. The two example neurons are in network module 1, and both prefer 90° with respect to cue 1. However, the congruent neuron prefers 90° of cue 2, while the opposite neuron prefers $-90$° with respect to cue 2. Error bar indicates the SD of firing rate across trials. (**B**) The neurometric function of the example congruent and opposite neuron in a discrimination task to determine whether the cue disparity $x_1 - x_2$ is larger than 0° or not. Lines are the cumulative Gaussian fit of the neurometric function. (**C**) Averaged neuronal discrimination thresholds of the example congruent and opposite neurons. Parameters: $\alpha_1 = 0.25 U_0$, $\alpha_2 = 0.8 U_0$, and others are the same as those in **Figure 4**.

DOI: https://doi.org/10.7554/eLife.43753.012

The following figure supplement is available for figure 8:

**Figure supplement 1.** Discrimination of heading direction by single neurons.

DOI: https://doi.org/10.7554/eLife.43753.013

stimuli. To resolve this challenge, we argue that the brain should carry out multisensory processing concurrently by employing congruent and opposite cells to realize complementary functions. Specifically, congruent neurons perform cue integration with opposite neurons computing the cue disparity simultaneously, so that the information they extract are complementary, based on which the neural system can assess the validity of integration and recover the lost information associated with single cues if necessary. Through this process, the brain can, on one hand, achieve improved stimulus perception if the cues are from the same stimulus of interest, and on the other hand, differentiate and recognize stimuli based on individual cues with little time delay if the cues are from different stimuli of interest. We built a biologically plausible network model to validate this processing strategy. The model consists of two reciprocally connected modules representing MSTd and VIP, respectively, and it carries out heading-direction inference based on visual and vestibular cues. Our model successfully reproduces the tuning properties of opposite neurons, verifying that opposite neurons encode the disparity information between cues, and demonstrates that the interplay between congruent and opposite neurons can implement concurrent multisensory processing.

Opposite neurons have been found in experiments for years (**Chen et al., 2013**; **Gu et al., 2008**), but their functional role remains a mystery. There have been few studies investigating this issue, and two computational works were reported (**Kim et al., 2016**; **Sasaki et al., 2017**), where the authors explored the contribution of opposite neurons in a computational task of inferring self-motion direction by eliminating the confound information of object motion. They showed that opposite neurons are essential, as they provide complementary information to congruent neurons necessary to accomplish the required computation. This result is consistent with our idea that opposite neurons are indispensable in multisensory processing, but our study goes one step further by theoretically proposing that opposite neurons encode the disparity information between cues and that congruent and opposite neurons jointly realize concurrent multisensory processing.

It is worthwhile to point out that in the present study, we have only demonstrated that congruent neurons implement Bayesian cue integration within the framework of a single-component prior and that opposite neurons encode the cue disparity information, and we have not explored whether they can combine together to realize a full Bayesian inference for multisensory processing. In the full Bayesian inference, also termed as the causal inference (**Körding et al., 2007**; **Sato et al., 2007**;

*Shams and Beierholm, 2010*), the neural system utilizes the prior knowledge about the probabilities of two cues coming from the same or different objects. The prior can be written as

$$p(s_1, s_2) = \sum_{C=1}^{2} p(s_1, s_2|C)p(C), \tag{9}$$

where $C = 1$ corresponds to the causal structure of two cues from the same object and $C = 2$ the causal structure of two cues from different objects. The posterior of stimuli is expressed as $p(s_1, s_2|x_1, x_2) = \sum_C p(s_1, s_2|x_1, x_2, C)p(C|x_1, x_2)$, which requires estimating the causal structure of cues. It is possible that opposite neurons, which encode the cue disparity information, can help the neural system to implement the causal inference. But to fully address this question, we need to resolve a number of issues, including the exact form of the prior, the network structure for realizing model selection, and the relevant experimental evidence, which will be the subject of our future research.

The present study only investigated integration and segregation of two sensory cues, but our model can be generalized to the cases of processing more than two cues that may happen in reality (*Wozny et al., 2008*). In such situations, the network model consists of $N > 2$ modules, and in module $m$, the received sensory cues can be differentiated as the direct one and the integrated results through combining all cues,

$$p_d(s_m|x_1, \ldots, x_N) \propto \frac{p(s_m|x_m)}{\left[\prod_{j=1}^{N} p(s_m|x_j)\right]^{1/N}}. \tag{10}$$

Congruent neurons can be reciprocally connected with each other between modules in the congruent manner as described above, so that they integrate the direct and all indirect cues optimally in the distributed manner. Opposite neurons could receive the direct cue from feedforward inputs (numerator in *Equation 10*), and receive the activities of congruent neurons in the opposite manner (denominator in *Equation 10*) through offset connection by 180°. The interplay between congruent and opposite neurons determines whether the direct cue should be integrated with all other cues at each module, and their joint activities can recover the stimulus information based only on the direct cue if necessary. This encoding strategy is similar with the norm-based encoding of face found in IT neurons (*Leopold et al., 2006*).

In the present study, we only demonstrated by analysis that the neural system can utilize the joint activities of congruent and opposite neurons to assess the validity of cue integration and to recover the information of direct cues in cue integration, but we did not go into the detail of how the brain actually carries out these operations. For assessing the validity of cue integration, essentially it is to compare the activities of congruent and opposite neurons and the winner indicates the choice. This competition process can be implemented easily in neural circuitry. For instance, it can be implemented by considering that congruent and opposite neurons are connected to the same inhibitory neuron pool which induces competition between them, such that only one group of neurons will sustain active responses after competition to represent the choice; alternatively, the activities of congruent and opposite neurons provide competing inputs to a decision-making network, and the latter generates the choice by accumulating evidence over time (*Wang, 2008*; *Engel and Wang, 2011*). Both mechanisms are feasible but further experiments are needed to clarify which one is used in practice. For recovering the stimulus information from direct cues by using the activities of congruent and opposite neurons, this study has shown that it can be done in a biologically plausible neural network, since the operation is expressed as solving the linear equation given by *Equation 8*. A concern is, however, whether recovering is really needed in practice, since at each module, the neural system may employ an additional group of neurons to retain the stimulus information estimated from the direct cue. An advantage of recovering the lost stimulus information by utilizing congruent and opposite neurons is saving the computational resource, but this needs to be verified by experiments.

The present study focused on investigating the role of opposite neurons in heading-direction inference with visual and vestibular cues as an example. In essence, the contribution of opposite neurons is to retain the disparity information between features to be integrated for the purpose of concurrent processing. We therefore expect that opposite neurons, or their counterparts of similar functions, is a general characteristic of neural information processing where feature integration and

segregation are involved (*Born, 2000*; *Thiele et al., 2002*; *Nadler et al., 2013*; *Goncalves and Welchman, 2017*). Indeed, for example, it has been found in the visual system, there exist 'what not' detectors which respond best to discrepancies between cues (analogous to opposite neurons) and they facilitate depth and shape perceptions (*Goncalves and Welchman, 2017*; *Rideaux and Welchman, 2018*). We hope that this study gives us insight into understanding the general principle of how the brain integrates/segregates multiple sources of information efficiently.

## Materials and methods

### Probabilistic model and its inference

The probabilistic model used in this study is widely adopted in multisensory research (*Bresciani et al., 2006*; *Ernst, 2006*; *Roach et al., 2006*; *Sato et al., 2007*). Suppose that two sensory cues $x_1$ and $x_2$ are independently generated by two underlying stimuli $s_1$ and $s_2$ respectively. In the example of visual-vestibular cue integration (*Fetsch et al., 2013*), $s_1$ and $s_2$ refer to the underlying visual and vestibular moving direction, while $x_1$ and $x_2$ are internal representations of moving direction in the visual and vestibular cortices. Because moving direction is a circular variable, we also assume that both $s_m$ and $x_m$ ($m = 1, 2$) are circular variables distributed in the range $(-\pi, \pi]$. Because each cue is independently generated by the corresponding underlying stimulus, the joint likelihood function can be factorized

$$p(x_1, x_2|s_1, s_2) = p(x_1|s_1)p(x_2|s_2).$$

In this study, each likelihood function $p(x_m|s_m)$ ($m = 1, 2$) is modeled by the von Mises distribution, which is a variant of circular Gaussian distribution (*Mardia and Jupp, 2009*; *Murray and Morgenstern, 2010*), given by *Equation 1*. Note that in *Equation 1*, $\kappa_m$ is a positive number characterizing the concentration of the distribution, which is analogous to the inverse of the variance ($\sigma^{-2}$) of Gaussian distribution. In the limit of large $\kappa_m$, a von Mises distribution $\mathcal{M}(x_m; s_m, \kappa_m)$ approaches to a Gaussian distribution with variance of $\kappa_m^{-1}$ (see details in Appendix 1, *Mardia and Jupp, 2009*).

The prior $p(s_1, s_2)$ specifies the probability of occurrence of $s_1$ and $s_2$, and is set as a von Mises distribution of the discrepancy between two stimuli (*Bresciani et al., 2006*; *Roach et al., 2006*; *Zhang et al., 2016*), given by *Equation 2*. Note that the marginal prior of either stimulus, for example $p(s_1) = \int_{-\pi}^{\pi} p(s_1, s_2) ds_2 = 1/2\pi$ is a uniform distribution.

### Inference

The inference of underlying stimuli can be conducted by using Bayes' theorem to derive the posterior

$$p(s_1, s_2|x_1, x_2) \propto p(x_1|s_1)p(x_2|s_2)p(s_1, s_2), \tag{11}$$

The posterior of either stimuli, for example stimulus $s_1$, can be obtained by marginalizing the joint posterior (*Equation 11*) as follows (the posterior of can be similarly obtained by interchanging indices 1 and 2)

$$
\begin{aligned}
p(s_1|x_1, x_2) &= \int_{-\pi}^{\pi} p(s_1, s_2|x_1, x_2) ds_2 \\
&\propto p(x_1|s_1) \int_{-\pi}^{\pi} p(x_2|s_2)p(s_1, s_2) ds_2 \\
&\propto p(s_1|x_1)p(s_1|x_2),
\end{aligned}
\tag{12}
$$

where we used the fact that both marginal distributions $p(s_m)$ and $p(x_m)$ are uniform and then interchanged the role of $x_m$ and $s_1$ in their conditional distributions. It indicates that the posterior of $s_1$ given two cues corresponds to a product of posterior of $s_1$ when each $x_m$ is individually presented, which could effectively accumulate the information of $s_1$ from both cues. $p(s_1|x_2)$ can be calculated as (see details in Appendix 1),

$$p(s_1|x_2) \propto \int_{-\pi}^{\pi} p(x_2|s_2)p(s_1, s_2) ds_2 \simeq \mathcal{M}(s_1; x_2, \kappa_{2s}), \text{ where } A(\kappa_{2s}) = A(\kappa_2)A(\kappa_s). \tag{13}$$

$A(\kappa) = \int_{-\pi}^{\pi} \cos\theta e^{\kappa\cos\theta} d\theta / \int_{-\pi}^{\pi} e^{\kappa\cos\theta} d\theta$ calculates the mean resultant length (first order trigonometric statistics), measuring the dispersion of a von Mises distribution. An approximation was used in the calculation through equating the mean resultant length of the integral with that of a von Mises distribution (*Mardia and Jupp, 2009*), because the integral of the product of two von Mises distributions is no longer a von Mises distribution. The meaning of $A(\kappa_{2s})$ can be understood by considering the Gaussian equivalent of von Mises distribution, where the inverse of concentration $\kappa^{-1}$ can approximate the variance of Gaussian distribution, yielding $\kappa_{2s}^{-1} \approx \kappa_2^{-1} + \kappa_s^{-1}$.

Finally, substituting the detailed expression into *Equation 12*,

$$
\begin{aligned}
p(s_1|x_1,x_2) &\propto \exp[\kappa_1\cos(s_1-x_1) + \kappa_{2s}\cos(s_1-x_2)] \\
&\propto \exp[(\kappa_1\cos x_1 + \kappa_{2s}\cos x_2)\cos s_1 + (\kappa_1\sin x_1 + \kappa_{2s}\sin x_2)\sin s_1] \\
&\propto \exp[\hat{\kappa}_1\cos(s_1-\hat{s}_1)].
\end{aligned}
$$

The expressions of the mean $\hat{s}_1$ and concentration $\hat{\kappa}_1$ can be found in *Equation 4*. The expressions of $\Delta\hat{s}_1$ and $\Delta\hat{\kappa}_1$ in the disparity information can be similarly calculated and is shown in *Equation 7*.

## Loss of cue information after integration

We could calculate the amount of cue information after integration in theory. Unlike the Gaussian distribution, it is not easy to analytically calculate the amount of information contained in a von Mises distribution. To simplify the analysis, we use a Gaussian approximation for a von Mises distribution first, and then calculate the amount of cue information contained in the posterior distribution $p(s_1, s_2|x_1, x_2)$ in Gaussian case. This approximation will significantly simplify the information analysis, without changing the basic conclusion and theoretical insight.

With a large concentration parameter $\kappa$, a von Mises distribution $\mathcal{M}(s; x, \kappa)$ can be approximated by a Gaussian distribution $\mathcal{N}(s; x, \kappa^{-1})$ (*Mardia and Jupp, 2009*). Thus, we approximate the von Mises likelihood $p(x_m|s_m) = \mathcal{M}(x_m; s_m, \kappa_m)$ into a Gaussian likelihood as $\mathcal{N}(x_m; s_m, \kappa_m^{-1})$, and approximate the von Mises prior $p(s_1, s_2)$ into a Gaussian prior as $\mathcal{N}(s_1; s_2, \kappa_s^{-1})$. Then the posterior distribution in the Gaussian case can be calculated to be (see *Zhang et al., 2016*),

$$
p(s|\mathbf{x}) = \mathcal{N}(s; \langle s|\mathbf{x}\rangle, \mathrm{Cov}(s|\mathbf{x})),
$$

where

$$
\begin{aligned}
\langle s|\mathbf{x}\rangle &= \begin{pmatrix} \kappa_2^{-1} + \kappa_s^{-1} & \kappa_1^{-1} \\ \kappa_2^{-1} & \kappa_1^{-1} + \kappa_s^{-1} \end{pmatrix} \begin{pmatrix} x_1 \\ x_2 \end{pmatrix}, \\
\mathrm{Cov}(s|\mathbf{x}) &= \frac{1}{\kappa_1^{-1} + \kappa_2^{-1} + \kappa_s^{-1}} \begin{pmatrix} \kappa_1^{-1}(\kappa_2^{-1} + \kappa_s^{-1}) & \kappa_1^{-1}\kappa_2^{-1} \\ \kappa_1^{-1}\kappa_2^{-1} & \kappa_2^{-1}(\kappa_1^{-1} + \kappa_s^{-1}) \end{pmatrix}.
\end{aligned}
$$

The Fisher information of cue $x_1$ contained in the posterior $p(s|\mathbf{x})$ can be calculated to be

$$
\begin{aligned}
\mathcal{I}(x_1)|_{p(s|\mathbf{x})} &= -\int \left[\frac{\partial^2}{\partial x_1^2}\ln p(s|\mathbf{x})\right] p(s|\mathbf{x}) ds \\
&= \frac{\partial\langle s|\mathbf{x}\rangle^\top}{\partial x_1} \mathrm{Cov}(s|\mathbf{x})^{-1} \frac{\partial\langle s|\mathbf{x}\rangle}{\partial x_1} \\
&= \kappa_1 \frac{\kappa_2^{-1} + \kappa_s^{-1}}{\kappa_1^{-1} + \kappa_2^{-1} + \kappa_s^{-1}}.
\end{aligned}
$$

The likelihood conveys all cue information, where the amount of information of cue $x_1$ in the likelihood is

$$
\mathcal{I}(x_1)|_{p(\mathbf{x}|s)} = \kappa_1.
$$

Thus the percentage of lost information of cue 1 is

$$
\begin{aligned}
Pct_{loss}(x_1) &= 1 - \frac{\mathcal{I}(x_1)|_{p(s|\mathbf{x})}}{\mathcal{I}(x_1)|_{p(\mathbf{x}|s)}} \\
&= \frac{\kappa_1^{-1}}{\kappa_1^{-1} + \kappa_2^{-1} + \kappa_s^{-1}}.
\end{aligned}
$$

We see the amount of information loss increases with $\kappa_s$, which controls the extent of integration (*Figure 1—figure supplement 1*). When $\kappa_s \to \infty$, the two cues will be fully integrated, and then the amount of information loss reaches maximum.

## Analysis leading to neural implementation

Here, we present the analysis that inspires us to propose the network model implementing integration and segregation.

### Neural encoding model

Suppose there is a population of $N$ neurons representing the estimate of stimulus $s_1$. We adopt a widely used encoding model that the firing activities $\mathbf{r}$ of neurons are independent with each other, and each satisfies a Poisson distribution with the rate specified by its tuning curve (*Ma et al., 2006*). In this encoding model for $s_1$ (the case for $s_2$ is similar),

$$
\begin{aligned}
\ln p(\mathbf{r}|s_1) \; &= \ln\left[\prod_{j=1}^{N} p(\mathbf{r}_j|s_1)\right] \\
&= \sum_{j=1}^{N} \ln\left[\frac{f_j(s_1)^{\mathbf{r}_j}}{\mathbf{r}_j!} e^{-f_j(s_1)}\right] \\
&= \sum_{j=1}^{N} \mathbf{r}_j f_j(s_1) - \sum_{j=1}^{N} f_i(s_1) - \sum_{j=1}^{N} \ln(\mathbf{r}_j!),
\end{aligned}
\tag{14}
$$

where $\mathbf{r}_j$ and $f_j(s_1)$ are the firing rate and tuning curve of $j$-th neuron representing $s_1$, respectively. Because heading direction is a circular variable ranging from $-\pi$ to $\pi$, the tuning curve can be modeled as a circular function,

$$
\begin{aligned}
f_j(s_1) \; &= f(\theta_j - s_1) \\
&= R\exp\left[a\cos(\theta_j - s_1)\right],
\end{aligned}
$$

where $R$ is the maximal firing rate of the neuron, $\theta_j$ is the preferred stimulus of $j$-th neuron, and the preference of all neurons $\{\theta_j\}_{j=1}^{N}$ uniformly cover the whole stimulus space. With the assumption that the summed mean firing rate of all neurons (the second term in *Equation 14*) is a constant irrelevant to stimulus value, and focusing on terms that are responsive to stimuli, we can get the detailed expression for the encoding model,

$$
\ln p(\mathbf{r}|s_1) = a\sum_{j=1}^{N} \mathbf{r}_j \cos(\theta_j - s_1) + const.
\tag{15}
$$

Then the distribution for stimulus $s_1$ becomes a von Mises distribution (*Mardia and Jupp, 2009*),

$$
p(s_1|\mathbf{r}) = \mathcal{M}(s_1; \hat{s}_1, \hat{\kappa}_1).
$$

The mean $\hat{s}_1$ and concentration $\hat{\kappa}_1$ of the stimulus are

$$
\begin{aligned}
\hat{s}_1 \; &= \tan^{-1}\left(\frac{\sum_{j=1}^{N} \mathbf{r}_j \sin\theta_j}{\sum_{j=1}^{N} \mathbf{r}_j \cos\theta_j}\right), \\
\hat{k}_1 \; &= \left[\left(\sum_{j=1}^{N} \mathbf{r}_j \sin\theta_i\right)^2 + \left(\sum_{j=1}^{N} \mathbf{r}_j \cos\theta_j\right)^2\right]^{1/2}.
\end{aligned}
$$

### Implementing multisensory integration

Given the encoding model, we then explore the neuronal operations required to implement multisensory integration given the neural representation mentioned above. Because the estimate of $s_1$ is fully represented by the neural population $\mathbf{r}$, the activities of the neural population that implements integration using *Equation (3)* should satisfy

$$
\ln p(s_1|\mathbf{r}(x_1, x_2)) = \ln p(s_1|\mathbf{r}(x_1)) + \ln p(s_1|\mathbf{r}(x_2)),
$$

where $\mathbf{r}(x_1, x_2)$ denotes the population firing activity given the cues $x_1$ and $x_2$ together, and similarly

for $\mathbf{r}(x_1)$ and $\mathbf{r}(x_2)$. Substituting the encoding model (*Equation 15*) into above equation, we can find that

$$\mathbf{r}_j(x_1, x_2) = \mathbf{r}_j(x_1) + \mathbf{r}_j(x_2).$$

The above equation indicates that the neuronal responses given two cues should be the combination of their responses when either cue is given, in order to implement integration. This is the same as the result in the previous work (*Ma et al., 2006*).

## Implementing multisensory segregation

Similarly, in order to implement multisensory segregation (*Equation 6*), the neuronal responses should satisfy

$$\ln p_d(s_1|\mathbf{r}(x_1, x_2)) = \ln p(s_1|\mathbf{r}(x_1)) - \ln p(s_1|\mathbf{r}(x_2)).$$

Substituting the neural encoding model into the above equation (*Equation 15*),

$$\sum_j \mathbf{r}_j(x_1, x_2) \cos(\theta_j - s_1) = \sum_j \mathbf{r}_j(x_1) \cos(\theta_j - s_1) - \sum_j \mathbf{r}_j(x_2) \cos(\theta_j - s_1).$$

At first sight, the above equation could indicate that the multisensory segregation can be achieved by the suppression from the neural activity when giving cue 2,

$$\mathbf{r}_j(x_1, x_2) = \mathbf{r}_j(x_1) - \mathbf{r}_j(x_2).$$

However, due to the constraint that the neuronal firing rate is a positive number, $\mathbf{r}_j(x_1, x_2)$ would be rectified to be zero if $\mathbf{r}_j(x_2)$ is larger than $\mathbf{r}_j(x_1)$. When this happens, the neurons fail to represent the magnitude of the disparity between two cues.

Fortunately, this problem can be resolved by using the property of cosine function that $\cos(x + \pi) = -\cos(x)$,

$$-\sum_j \mathbf{r}_j(x_2) \cos(\theta_j - s_1) = \sum_j \mathbf{r}_j(x_2) \cos[\underbrace{(\theta_j + \pi)}_{\theta_{j'}} - s_1)]$$
$$= \sum_j \mathbf{r}_{j'}(x_2) \cos(\theta_j - s_1), \quad \text{where } \theta_j = \theta_{j'} + \pi.$$

The second equality is obtained through changing the dummy variables $j$ and $j'$. Canceling the cosine terms, it can be derived that the activity of each neuron should satisfy

$$\mathbf{r}_j(x_1, x_2) = \mathbf{r}_j(x_1) + \mathbf{r}_{j'}(x_2), \text{ where } \theta_j = \theta_{j'} + \pi.$$

The above equation indicates that in order to achieve optimal segregation, the neurons should combine the neuronal responses under direct cue $\mathbf{r}_j(x_1)$, and the responses under indirect cue but rotated to the opposite direction $\mathbf{r}_{j'}(x_2)$. This is consistent with the definition of opposite neurons (*Gu et al., 2008*; *Chen et al., 2013*).

## Dynamics of a decentralized network model

We adopted a decentralized network model to implement concurrent multisensory integration and segregation (*Zhang et al., 2016*). The network model is composed of two modules, with each module consisting of two groups of neurons with the same number: one is intended to model congruent neurons and another is for opposite neurons. Each neuronal group is modeled as a continuous attractor neural network (*Wu et al., 2008*; *Fung et al., 2010*; *Zhang and Wu, 2012*), which has been widely used to model the coding of continuous stimuli in the brain (*Ben-Yishai et al., 1995*; *Georgopoulos et al., 1986*; *Samsonovich and McNaughton, 1997*) and it can optimally implement maximal likelihood inference (*Deneve et al., 1999*; *Wu et al., 2002*). Denote $u_m^n(\theta, t)$ and $r_m^n(\theta, t)$ as the synaptic input and firing rate at time $t$ respectively for an $n$-type neuron ($n = c, o$ represents the congruent and opposite neurons, respectively) in module $m$ ($m = 1, 2$) whose preferred heading direction with respect to the feedforward cue $m$ is $\theta$. It is worthwhile to emphasize that $\theta$ is the preferred direction only to the feedforward cue, for example the feedforward cue to network module 1

is cue 1, but $\theta$ does not refer to the preferred direction given another cue, because the preferred direction of an opposite neuron given each cue is different. In the network model, the network module m = 1, 2 can be regarded as the brain areas MSTd and VIP, respectively. For simplicity, we assume that the two network modules are symmetric, and only present the dynamical equations for network module 1. The dynamical equations for network module 2 can be obtained by interchanging the indices 1 and 2 in the following dynamical equations.

The dynamics of the synaptic input of $n$-type neurons in network module $m$, $u_m^n(\theta, t)$, is governed by

$$\tau \frac{\partial u_m^n(\theta, t)}{\partial t} = -u_m^n(\theta, t) + \sum_{\theta'=-\pi}^{\pi} W_{rc}(\theta, \theta') r_m^n(\theta', t) + \sum_{\theta'=-\pi}^{\pi} W_{rp}^n(\theta, \theta') r_{k \neq m}^n(\theta', t) + I_m^n(\theta, t),$$  (16)

where $I_m^n(\theta, t)$ is the feedforward inputs from unisensory brain areas conveying cue information. $W_{rc}(\theta, \theta')$ is the recurrent connections from neuron $\theta'$ to neuron $\theta$ within the same group of neurons and in the same network module, which is set to be

$$W_{rc}(\theta, \theta') = \frac{J_{rc}}{2\pi I_0(a)} \exp[a \cos(\theta - \theta')],$$  (17)

where $a$ is the connection width and effectively controls the width of neuronal tuning curves. $W_{rp}^n(\theta, \theta')$ denotes the reciprocal connections between congruent neurons across network modules ($n = c$), or between opposite neurons across network modules ($n = o$). $W_{rp}^c(\theta, \theta')$ is the reciprocal connections between congruent cells across two modules (the superscript $c$ denotes the connections are in a congruent manner, that is a 0° neuron will have the strongest connection with a 0° neuron),

$$W_{rp}^c(\theta, \theta') = \frac{J_{rp}}{2\pi I_0(a)} \exp[a \cos(\theta - \theta')].$$  (18)

Note that $\theta$ and $\theta'$ in the above equation denote the preferred direction of two neurons at different network modules over their respective feedforward cues. For simplicity, $W_{rp}^c(\theta, \theta')$ and $W_{rc}(\theta, \theta')$ have the same connection width $a$. This simplification does not change the basic conclusion substantially. A previous study indicates that the reciprocal connection strength $J_{rp}$ determines the extent of cue integration, and effectively represents the correlation of two underlying stimuli in the prior $p(s_1, s_2)$ (Zhang et al., 2016). Moreover, the opposite neurons from different network modules are connected in an opposite manner with an offset of $\pi$,

$$W_{rp}^o(\theta, \theta') = \frac{J_{rp}}{2\pi I_0(a)} \exp[a \cos(\theta - \theta' + \pi)].$$  (19)

Hence, an opposite neurons preferring 0° of cue 1 in network module 1 will have the strongest connection with the opposite neurons preferring of 180° of cue 2 in network module 2. It is worthwhile to note that the strength and width of $W_{rp}^c(\theta, \theta')$ and $W_{rp}^o(\theta, \theta')$ are the same, in order to convey the same information from the indirect cue. This is also supported by the fact that the tuning curves of the congruent and opposite neurons have similar tuning strengths and widths (Chen et al., 2011).

Each neuronal group contains an inhibitory neuron pool which sums all excitatory neurons' activities and then divisively normalize the response of the excitatory neurons,

$$r_m^n(\theta, t) = \frac{[u_m^n(\theta, t)]_+^2}{1 + \omega D_m^n(t)},$$  (20)

where $\omega$ controls the magnitude of divisive normalization, and $[x]_+ = \max(x, 0)$ is the negative rectified function. $D_m^n(t)$ denotes the response of the inhibitory neuron pool associated with neurons of type $n$ in network module $m$ at time $t$, which sums up the synaptic inputs of the same type of excitatory neurons $u_m^n(\theta, t)$ and also receives the inputs from the other type of neurons $u_m^{n'}(\theta, t)$,

$$D_m^n(t) = \sum_\theta [u_m^n(\theta, t)]_+^2 + J_{int} \sum_\theta [u_m^{n'}(\theta, t)]_+^2.$$  (21)

$J_{int}$ is a positive coefficient not larger than 1, which effectively controls the sharing between the inhibitory neuron pool associated with the congruent and opposite neurons in the same network

module. The partial share of the two inhibitory neuron pools inside the same network module introduces competition between two types of neurons, improving the robustness of network.

The feedforward inputs convey the direct cue information from the unisensory brain area to a network module, for example the feedforward inputs received by MSTd neurons is from MT which extracts the heading direction from optic flow,

$$I_m^n(\theta, t) = I_m^{ff}(\theta) + \sqrt{FI_m^{ff}(\theta)}\xi_m(\theta, t) + I_b + \sqrt{FI_b}\epsilon_m^n(\theta, t),$$

$$\text{where} \quad I_m^{ff}(\theta) = \alpha_m \exp[a\cos(\theta - x_m)/2 - a/2]. \tag{22}$$

The feedforward inputs contain two parts: one conveys the cue information (the first two terms in above equation) and another the background inputs (the last two terms in the above equation), which are always present no matter whether a cue is presented or not. The variance of the noise in the feedforward inputs $FI_m^{ff}(\theta)$ is proportional to their mean, and $F$ characterizes the Fano factor. The multiplicative noise is in accordance with the Poisson variability of the cortical neurons' response. $\alpha_m$ is the intensity of the feedforward input and effectively controls the reliability of cue $m$. $x_m$ is the direction of cue $m$. $I_b$ is the mean of background input. $\xi_m(\theta, t)$ and $\epsilon_m^n(\theta, t)$ are mutually independent Gaussian white noises of zero mean with variances satisfying $\langle \xi_m(\theta, t)\xi_{m'}(\theta', t') \rangle = \delta_{mm'}\delta(\theta - \theta')\delta(t - t')$, and $\langle \epsilon_m^n(\theta, t)\epsilon_{m'}^{n'}(\theta', t') \rangle = \delta_{mm'}\delta_{nn'}\delta(\theta - \theta')\delta(t - t')$. Note that the cue-associated noise $\xi_m(\theta, t)$ to congruent and opposite neurons are exactly the same, while the background noise $\epsilon_m^n(\theta, t)$ to congruent and opposite neurons are independent of each other. Previous works indicated that the exact form of the feedforward inputs is not crucial, as long as they have a uni-modal shape (*Zhang and Wu, 2012*).

## Network simulation and parameters

In our simulation, each network module contains 180 congruent and opposite neurons, respectively, whose preferred direction with respect to the feedforward cue is uniformly distributed in the feature space $(-180°, 180°]$. For simplicity, the parameters of the two network modules were chosen symmetric with each other, that is all structural parameters of the two modules have the same value. The synaptic time constant $\tau$ was rescaled to one as a dimensionless number and the time step size was $0.01\tau$ in simulation. All connections have the same width $a = 3$, which is equivalent to a value of about 40° for the width of tuning curves of the neurons. The dynamical equations are solved by using Euler method.

The range of parameters was listed in the following if not mentioned otherwise. The detailed parameters for each figure can be found in figure captions. The strength of divisive normalization was $\omega = 3 \times 10^{-4}$, and $J_{int} = 0.5$ which controls the proportion of share between the inhibition pools affiliated with congruent and opposite neurons in the same module (*Equation 21*). The absolute values of $\omega$ and $J_{int}$ did not affect our basic results substantially, and they only determine the maximal firing rate the neurons can reach. Of the particular values we chose, the firing rate of the neurons saturates at around 50 Hz. The recurrent connection strength between neurons of the same type and in the same network module was $J_{rc} = [0.3, 0.4]J_c$, where $J_c$ is the minimal recurrent strength for a network module to hold persistent activity after switching off feedforward inputs. The expression of $J_c$ is shown in *Equation (A39)* in Appendix 3. The strength of the reciprocal connections between the network modules is $J_{rp} = [0.1, 0.9]J_{rc}$, and is always smaller than the recurrent connection strength within the same network module. The sum of the recurrent strength $J_{rc}$ and reciprocal strength $J_{rp}$ cannot be too large, since otherwise the congruent and opposite neurons in the same network module will have strong competition resulting in the emergence of winner-take-all behavior. However, the winner-take-all behavior was not observed in experiments. The input intensity $\alpha$ was scaled relative to $U_0 = J_c e^{a/2}/[2\pi\omega(1 + J_{int})I_0(a/2)]$, and is distributed in $[0.3, 1.5]U_0$, where $U_0$ is the value of the synaptic bump height that a group of neurons can hold without receiving feedforward input and reciprocal inputs when $J_{rc} = J_c$. The range of the input intensity was chosen to be wide enough to cover the super-linear to nearly saturated regions of the input-firing rate curve of the neurons. The strength of the background input was $I_b = 1$, and the Fano factors of feedforward and background inputs were set to 0.5, which led to the Fano factor of single neuron responses taking values of the order 1. In simulations, the position of the population activity bump was read out by calculating the

population vector (*Georgopoulos et al., 1986*; *Dayan and Abbott, 2001*). For example, the position of the population activities of the congruent neurons in module 1 at time $t$ was estimated as

$$z_1^c(t) = \arg\left[\sum_\theta r_1^c(\theta, t)e^{j\theta}\right], \tag{23}$$

where $j$ is the imaginary unit, and the function $\arg[\cdot]$ outputs the angle of a vector. Note that $\theta$ is the preferred direction over the direct cue conveyed by feedforward inputs. For the example pertaining to the above equation, $\theta$ refers to the preference over cue 1. To reproduce the tuning curves (*Figures 5* and *6*), the network dynamics was simulated for a single long trial and the neuronal responses in equilibrium state was averaged over time to get the mean and concentration of the firing rate distribution. To perform ROC analysis (*Figure 8* and *Figure 8—figure supplement 1*), the network model was simulated for 30 trials. The number of trials is consistent with experimental studies (*Gu et al., 2008*), and it does not influence the results substantially as long as it is large enough. The network model was simulated by using MATLAB, and the corresponding code can be found at https://github.com/wenhao-z/Opposite_neuron (copy archived at https://github.com/elifesciences-publications/Opposite_neuron).

## Demo tasks of network model

### Testing network's performance of integration and segregation

We compared the network's estimate under three cueing conditions in simulations, that is either cue 1 or cue 2 is individually presented, or both cues are simultaneously presented. In each cueing condition, we simulate the network dynamics for sufficient long time to guarantee it is in equilibrium state, where the estimates made by congruent and opposite neurons in the two network modules are decoded respectively. Denote $z_m^n(t|x_l)$ as the bump position at time $t$ when only cue $x_l$ ($l = 1, 2$) is presented. Simulations show that the distribution of the bump position over time is well approximated by a von Mises distribution. The mean of the estimate is obtained through averaging across time (equivalent to average across trials at equilibrium) (*Mardia and Jupp, 2009*),

$$\langle z_m^n|x_l\rangle = \arg\left(\frac{1}{N_t}\sum_t e^{jz_m^n(t|x_l)}\right),$$

where $N_t$ denotes the number of data points and is set to $5 \times 10^4$ in simulation. To estimate the concentration of the probabilistic population code, we consider the posterior distribution of the population vector decoded from each individual instance, rather than the width distribution of the bumps obtained from the individual instances. Hence we consider the mean resultant length of the von Mises distribution given by *Equation (A4)*. When the distribution is sufficiently sharp, it can be approximated by the von Mises distribution in the neighborhood of the peak. Hence the concentration is estimated by

$$\kappa(z_m^n|x_l) = A^{-1}\left(\left|\frac{1}{N_t}\sum_t e^{jz_m^n(t|x_l)}\right|\right),$$

where $A^{-1}(\cdot)$ denotes the inverse function of $A(\cdot)$ in *Equation (A4)*. To verify whether the congruent neurons in each module achieve optimal cue integration, we calculated the theoretical prediction obtained by adding the estimates of the same group of neurons in *single cue* conditions according to *Equation (4)* (corresponding to the sum of the green vectors in *Figure 3B*),

$$\tilde{\kappa}_m^c e^{j\tilde{z}_m^c} = \sum_{l=1}^2 \kappa(z_m^c|x_l)e^{j\langle z_m^c|x_l\rangle},$$

where $\tilde{z}_m^c$ and $\tilde{\kappa}_m^c$ denote, respectively, the predicted mean and concentration for the estimate of congruent neurons in module $m$ in the *combined cueing* condition. This prediction is then compared with the *actual* mean and concentration of the estimate from the same group of neurons in the *combined cueing* condition. Results are displayed in *Figure 6—figure supplement 1*.

We further tested whether the opposite neurons in a module implements optimal cue segregation. The theoretical prediction was obtained by substituting the mean and concentration of the

posterior represented by congruent neurons under single cue conditions into *Equation (7)* (corresponding to the difference of the green vectors in *Figure 3B*),

$$\tilde{\kappa}_m^o e^{j\tilde{z}_m^o} = \kappa(z_m^c|x_m)e^{j\langle z_m^c|x_m\rangle} - \kappa(z_m^c|x_{m'})e^{j\langle z_m^c|x_{m'}\rangle}, \tag{24}$$

where $\tilde{z}_m^o$ and $\tilde{\kappa}_m^o$ denote, respectively, the predicted mean and concentration of the estimate of opposite neurons in module $m$ in the combined cue condition. It is expected that the estimates of congruent and opposite neurons have the same mean and concentration given the direct cue, that is $\kappa(z_m^c|x_m)e^{j\langle z_m^c|x_m\rangle} = \kappa(z_m^o|x_m)e^{j\langle z_m^o|x_m\rangle}$, while given the indirect cue, their estimates have the same concentration but opposite mean, that is $\kappa(z_m^c|x_{m'})e^{j\langle z_m^c|x_{m'}\rangle} = -\kappa(z_m^o|x_{m'})e^{j\langle z_m^o|x_{m'}\rangle}$. Thus, the theoretical prediction for opposite neurons can also be obtained by

$$\tilde{\kappa}_m^o e^{j\tilde{z}_m^o} = \kappa(z_m^o|x_m)e^{j\langle z_m^o|x_m\rangle} + \kappa(z_m^o|x_{m'})e^{j\langle z_m^o|x_{m'}\rangle}. \tag{25}$$

We checked that *Equations (24, 25)* give the same prediction on the estimate of the opposite neurons. We used *Equation (25)* to predict the estimate of the opposite neurons in the combined cue condition. Results are presented in *Figure 6—figure supplement 1*.

## Reconstructing stimulus estimate under direct cue from congruent and opposite neurons' activity

The stimulus estimate from its direct cue can be recovered from the joint activities of congruent and opposite neurons in real-time when two cues are simultaneously presented. *Equation 8* indicates that the reconstruction of the posterior distribution of the direct cue can be achieved by multiplying the decoded distribution from congruent and opposite neurons in a network module. Thus, for example, the reconstructed estimate of stimulus one at time $t$ given its direct cue can be obtained by

$$\hat{s}_1(t)|x_1 = \arg\left[\left(\textstyle\sum_\theta r_1^c(\theta,t)\right)e^{jz_1^c(t)} + \left(\textstyle\sum_\theta r_1^o(\theta,t)\right)e^{jz_1^o(t)}\right], \tag{26}$$

where $z_1^c(t)$ and $z_1^o(t)$ are the positions of the population activities of the congruent and opposite neurons in network module 1, respectively, which were decoded by using population vector (*Equation 23*). In real-time reconstruction, the sum of firing rate represents the concentration of the distribution. This is supported by the finding that the reliability of the distribution is encoded by the summed firing rate in probabilistic population code (*Ma et al., 2006*; *Zhang et al., 2016*).

## Discriminating cue disparity on single neurons

A discrimination task was designed on the responses of single neurons to demonstrate that opposite neurons encode cue disparity information. The task is to discriminate whether the cue disparity, $x_1 - x_2$, is either smaller or larger than 0°. In the discrimination task, the mean direction of two cues, $x_1 + x_2 = 0$, is fixed at 0°, in order to rule out the influence of the change of integrated direction to neuronal activity. Meanwhile, the disparity between two cues, $x_1 - x_2$, is changed from −32° to 32° with a step of 4°. For each combination of cue direction, we applied three cueing conditions (cue 1, cue 2, combined cues) to the network model for 30 trials and the firing rate distributions of the single neurons were obtained (*Figure 8A and B*).

We chose an example congruent neuron preferring 90° in network module 1, and also an example opposite neuron in network module 1 preferring 90° with respect to cue 1. We used receiver operating characteristic (ROC) analysis (*Britten et al., 1992*) to compute the discriminating ability of the example neurons on cue disparity. The ROC value counts the proportion of instances where the direction of cue 1, $x_1$, is larger than the one of cue 2. Neurometric functions (*Figure 8B and E*) were constructed from those ROC values and were fitted with cumulative Gaussian functions by least square, and then the standard deviation of the cumulative Gaussian function was interpreted as the neuronal discrimination threshold (*Figure 8C*) (*Gu et al., 2008*). A smaller value of the discrimination threshold means that the neuron is more sensitive in the discrimination task. Although we adopted the von Mises distribution in the probabilistic model, the firing rate distribution of single neurons can be well fitted by a Gaussian distribution, justifying the use of the cumulative Gaussian distribution to fit the ROC values.

## Discriminating heading direction on single neurons

To reproduce experimental findings (*Gu et al., 2008*; *Chen et al., 2013*), we conducted a task of discriminating whether a stimulus value is smaller or larger than 0° based on the activities of an example congruent and an opposite neurons which are the same as the one described in Materials and methods. The directions of the two cues were always the same, and were simultaneously changed from −32° to 32°. The construction of neurometric function and the estimate of neuronal discrimination threshold are the same as the discrimination task presented in main text.

Similar with typical cue experiments (*Chen et al., 2013*; *Gu et al., 2008*), for each neuron, we used the Gaussian distribution to predict the discrimination threshold under combined cues by those under separate single-cue conditions,

$$\sigma_{\text{prediction}} = \sigma_1 \sigma_2 / \sqrt{\sigma_1^2 + \sigma_2^2}, \tag{27}$$

where $\sigma_1$ and $\sigma_2$ are the neuronal discrimination thresholds of a neuron under cue 1 and cue 2 conditions, respectively. The results are presented in *Figure 8—figure supplement 1*.

## Acknowledgements

This work is supported by Research Grants Council of Hong Kong (N_HKUST606/12, 605813, 16322616, and 16306817), National Basic Research Program of China (2014CB846101), Natural Science Foundation of China (31261160495), NSF 1816568 and IARPA contract D16PC00007.

## Additional information

### Funding

| Funder | Grant reference number | Author |
| --- | --- | --- |
| Research Grants Council, University Grants Committee | N_HKUST606/12 | KY Michael Wong |
| Research Grants Council, University Grants Committee | 605813 | KY Michael Wong |
| Research Grants Council, University Grants Committee | 16322616 | KY Michael Wong |
| Research Grants Council, University Grants Committee | 16306817 | KY Michael Wong |
| National Basic Research Program of China | 2014CB846101 | Si Wu |
| Natural Science Foundation of China | 31261160495 | Si Wu |
| National Science Foundation | 1816568 | Tai Sing Lee |
| Intelligence Advanced Research Projects Activity | D16PC00007 | Tai Sing Lee |

The funders had no role in study design, data collection and interpretation, or the decision to submit the work for publication.

### Author contributions

Wen-Hao Zhang, He Wang, KY Michael Wong, Si Wu, Conceptualization, Formal analysis, Writing—original draft, Writing—review and editing; Aihua Chen, Yong Gu, Resources, Validation; Tai Sing Lee, Formal analysis

### Author ORCIDs

Wen-Hao Zhang https://orcid.org/0000-0001-7641-5024
He Wang http://orcid.org/0000-0003-2101-8683

Yong Gu [iD] http://orcid.org/0000-0003-4437-8956
KY Michael Wong [iD] https://orcid.org/0000-0002-3078-4577

**Decision letter and Author response**
Decision letter https://doi.org/10.7554/eLife.43753.019
Author response https://doi.org/10.7554/eLife.43753.020

## Additional files

**Supplementary files**
• Transparent reporting form
DOI: https://doi.org/10.7554/eLife.43753.014

**Data availability**

The submitted manuscript presents a theoretical network modelling work. All codes used in this study has been uploaded to GitHub (https://github.com/wenhao-z/Opposite_neuron; copy archived at https://github.com/elifesciences-publications/Opposite_neuron) and can be openly accessed.

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

## Appendix 1

DOI: https://doi.org/10.7554/eLife.43753.015

# Background of the von Mises Distribution

## Definition of the von Mises distribution

The von Mises probability density function for a circular variable $x$ is defined as

$$\mathcal{M}(x;\mu,\kappa) = \frac{1}{2\pi I_0(\kappa)}\exp[\kappa\cos(x-\mu)], \tag{A1}$$

where $\mu$ is the mean of $x$, and the concentration parameter $\kappa$ measures the dispersion of $x$ around its mean value. $I_0(\kappa)$ is the modified Bessel function of the first kind and zero order, which is given by

$$I_0(\kappa) = \frac{1}{2\pi}\int_0^{2\pi}\exp(\kappa\cos x)dx. \tag{A2}$$

Note that $\mathcal{M}(x;\mu-\pi,\kappa)$ is equal to $\mathcal{M}(x;\mu,-\kappa)$. To avoid the indeterminacy of the parameter $\kappa$, it is usual to take $\kappa>0$.

Apart from using $\kappa$ to measure the concentration, we usually use the *mean resultant length* $\rho$ to measure the dispersion of a circular variable, because it can be more easily estimated from sampled data. The mean resultant length is defined as

$$\rho = \mathbb{E}[\cos(x-\mu)]. \tag{A3}$$

Note that $0 \leq \rho \leq 1$ means that the distribution is fully concentrated at the point $\mu$, while $\rho = 0$ means that the distribution is so scattered that there is no concentration around any particular point.

For a von Mises distribution with $\mu = 0$, its mean resultant length is calculated to be

$$\begin{aligned}\rho \ &\equiv A(\kappa) \\ &= \frac{1}{2\pi I_0(\kappa)}\int_0^{2\pi}\cos(x)e^{\kappa\cos x}dx.\end{aligned} \tag{A4}$$

## Relationship to the normal distribution

When $\kappa$ is large, we let $\xi = \kappa^{1/2}(x-\mu)$, and the von Mises distribution is approximated to be

$$\mathcal{M}(\xi;0,\kappa) \propto \exp\left(-\kappa[1-\cos(\kappa^{-1/2}\xi)]\right). \tag{A5}$$

Further approximating $1 - \cos(\kappa^{-1/2}\xi) = \frac{1}{2}\kappa^{-1}\xi^2 + \mathcal{O}(\kappa^{-2})$ for small $\xi$, we have

$$\mathcal{M}(\xi;0,\kappa) \propto \exp\left(-\xi^2/2\right) \propto \mathcal{N}(\xi;0,1). \tag{A6}$$

Thus, the von Mises distribution can be approximated to be a normal distribution for large $\kappa$ and small $|x-\mu|$, that is

$$\mathcal{M}(x;\mu,\kappa) \approx \mathcal{N}(x;\mu,\kappa^{-1}). \tag{A7}$$

## Relationship to the wrapped normal distribution

In general, a von Mises distribution can be approximated by a wrapped normal distribution with the same mean $\mu$ and the same mean resultant length $A(\kappa)$. The wrapped normal distribution $\mathcal{WN}(x;\mu,\rho)$ is obtained by wrapping a normal distribution on a circle. For a

random variable $x$, the corresponding random variable $x_w$ of the wrapped distribution is obtained by

$$x_w = x \pmod{2\pi}, \tag{A8}$$

and the wrapped distribution satisfies

$$f_w(x) = \sum_{k=-\infty}^{\infty} f(x + 2k\pi), \tag{A9}$$

where $f(x)$ is the probability density function of $x$.

Hence the probability density function of the wrapped normal distribution is defined as

$$\mathcal{WN}(x; \mu, \rho) = \frac{1}{\sqrt{2\pi}\sigma} \sum_{k=-\infty}^{\infty} \exp\left[-\frac{(x - \mu + 2k\pi)^2}{2\sigma^2}\right], \tag{A10}$$

where $\rho = \exp(-\sigma^2/2)$ is mean resultant length of the wrapped normal distribution.

By matching the mean and the mean resultant length of a von Mises distribution and a wrapped normal distribution, we have following approximation,

$$\mathcal{M}(x; \mu, \kappa) \simeq \mathcal{WN}(x; \mu, A(\kappa)) + \mathcal{O}(\kappa^{-1/2}), \quad \kappa \to \infty. \tag{A11}$$

It has been shown that this approximation works very well, even in the worst case when $\kappa \sim 1.4$ (ch. 3 in **Mardia and Jupp, 2009**).

## Product of two von Mises distributions

The cue integration involves calculating the product of two von Mises distributions (see **Equation 3** in the main text)

$$p(s|x_1, x_2) \propto p(x_1|s)p(x_2|s), \tag{A12}$$

where $p(x_m|s) = \mathcal{M}(s; x_m, \kappa_m)$ for $m = 1, 2$. Substituting detailed expressions, the right hand side of the above equation is,

$$p(s|x_1)p(s|x_2) = \frac{1}{(2\pi)^2 I_0(\kappa_1) I_0(\kappa_2)} \exp[\kappa_1 \cos(s - x_1) + \kappa_2 \cos(s - x_2)]. \tag{A13}$$

The two cosine terms inside the exponential function in the above equation can be merged together,

$$\begin{aligned}
&\kappa_1 \cos(s - x_1) + \kappa_2 \cos(s - x_2) \\
&= \kappa_1(\cos x_1 \cos s + \sin x_1 \sin s) + \kappa_2(\cos x_2 \cos s + \sin x_2 \sin s) \\
&= (\kappa_1 \cos x_1 + \kappa_2 \cos x_2) \cos s + (\kappa_1 \sin x_1 + \kappa_2 \sin x_2) \sin s \\
&= \kappa_3 \cos(s - x_3),
\end{aligned} \tag{A14}$$

where

$$\begin{aligned}
\kappa_3 &= \left[(\kappa_1 \cos x_1 + \kappa_2 \cos x_2)^2 + (\kappa_1 \sin x_1 + \kappa_2 \sin x_2)^2\right]^{1/2} \\
&= \left[\kappa_1^2 + \kappa_2^2 + 2\kappa_1\kappa_2 \cos(x_1 - x_2)\right]^{1/2},
\end{aligned} \tag{A15}$$

$$x_3 = \tan^{-1}\left(\frac{\kappa_1 \sin x_1 + \kappa_2 \sin x_2}{\kappa_1 \cos x_1 + \kappa_2 \cos x_2}\right). \tag{A16}$$

It is worthwhile to note that **Equations. (A15 and A16)** can be concisely expressed in complex representation,

$$\kappa_3 e^{jx_3} = \kappa_1 e^{jx_1} + \kappa_2 e^{jx_2}, \tag{A17}$$

where $\kappa e^{jx}$ geometrically corresponds to a vector in the 2D complex plane, with $\kappa$ and $x$ representing the length and angle of the vector, respectively.

Adding the normalization constant, we get

$$p(s|x_1, x_2) = \frac{1}{2\pi I_0(\kappa_3)} \exp[\kappa_3 \cos(s - x_3)]. \tag{A18}$$

## Integral of the product of two von Mises distributions

The calculation of $p(x_2|s_1)$ involves the integral of the product of two von Mises distributions,

$$
\begin{aligned}
p(x_2|s_1) &= \int_0^{2\pi} p(x_2|s_2) p(s_2|s_1) ds_2 \\
&= \frac{1}{(2\pi)^2 I_0(\kappa_1) I_0(\kappa_s)} \int_0^{2\pi} \exp[\kappa_2 \cos(s_2 - x_2) + \kappa_s \cos(s_2 - s_1)] ds_2.
\end{aligned} \tag{A19}
$$

Using the results in **Equations (A14-A16)**, we get

$$p(x_2|s_1) = \frac{I_0\left( \left[\kappa_2^2 + \kappa_s^2 + 2\kappa_2 \kappa_s \cos(s_1 - x_2)\right]^{1/2} \right)}{2\pi I_0(\kappa_2) I_0(\kappa_s)}. \tag{A20}$$

The above equation is not a von Mises distribution, but it can be approximated as one. The two von Mises distributions in **Equation (A19)** can be approximated by wrapped normal distributions, respectively (see **Equation A11**), which are

$$p(x_2|s_2) = \mathcal{M}(x_2; s_2, \kappa_2) \simeq \mathcal{WN}(s_2; x_2, A(\kappa_2)), \tag{A21}$$

$$p(s_2|s_1) = \mathcal{M}(s_2; s_1, \kappa_s) \simeq \mathcal{WN}(s_2; s_1, A(\kappa_s)). \tag{A22}$$

With these approximations, **Equation (A19)** becomes

$$
\begin{aligned}
p(x_2|s_1) &\simeq \int_0^{2\pi} \mathcal{WN}(x_2; s_2, A(\kappa_2)) \mathcal{WN}(s_2; s_1, A(\kappa_s)) ds_2 \\
&= \mathcal{WN}(x_2; s_1, A(\kappa_2) A(\kappa_2)).
\end{aligned} \tag{A23}
$$

Using the approximation of **Equation (A11)**, we finally get

$$p(x_2|s_1) \simeq \mathcal{M}\left(x_2; s_1, A^{-1}\{A(\kappa_2) A(\kappa_s)\}\right). \tag{A24}$$

# Appendix 2

DOI: https://doi.org/10.7554/eLife.43753.015

## Multisensory integration with Gaussian distribution

In the main text, we came across the probabilistic multisensory integration with von Mises distributions. To see its difference with that using Gaussian distribution, we present the result for Gaussian distribution below. In the Gaussian case, the likelihood function is given by

$$p(x_m|s_m) = \mathcal{N}(x_m; s_m, \sigma_m^2) = \frac{1}{\sqrt{2\pi}\sigma_m}\exp\left[-\frac{(x_m - s_m)^2}{2\sigma_m^2}\right], \tag{A25}$$

where the inverse of the variance of Gaussian distribution is related to the concentration of von Mises distribution (*Equation 1*), that is $\sigma_m^{-2} \approx \kappa_m$, for large $\kappa_m$ (*Equation A7*).

The stimulus prior in Gaussian distribution is written as (compared to *Equation 2*),

$$p(s_1, s_2) = \frac{1}{\sqrt{2\pi}\sigma_s L_s}\exp\left[-\frac{(s_1 - s_2)^2}{2\sigma_s^2}\right], \tag{A26}$$

where $L_s = 2\pi$ for heading direction.

Substituting *Equations (A25 and A26)* into *Equation (3)*, the posterior $p(s_1|x_1, x_2)$ is calculated to be

$$p(s_1|x_1, x_2) = \mathcal{N}(s_1; \hat{s}_1, \hat{\sigma}_1^2), \tag{A27}$$

where the mean and variance of the posterior are

$$\hat{\sigma}_1^{-2} = \sigma_1^{-2} + (\sigma_2^2 + \sigma_s^2)^{-1}, \tag{A28}$$

$$\hat{s}_1 = \hat{\sigma}_1^2\left[\sigma_1^{-2}x_1 + (\sigma_2^2 + \sigma_s^2)^{-1}x_2\right], \tag{A29}$$

Note that the reliability of cue integration using von Mises distribution decreases with the cue disparity $(x_1 - x_2)$ (see *Equation A15*), but in the Gaussian case, the reliability of cue integration $\hat{\sigma}_1^{-2}$ is independent of the cue disparity.

## Appendix 3

DOI: https://doi.org/10.7554/eLife.43753.015

### Theoretical analysis of a single network module

We conduct theoretical analysis to understand the dynamics of a single network module without receiving feedforward inputs and reciprocal inputs from another module. This analysis could help us to understand how recurrent connections between neurons and the divisive normalization determine the neural dynamics, and help us to set network parameters.

Cutting off feedforward and reciprocal inputs corresponds to setting $I_m^n(\theta, t) = 0$ and $J_{rp} = 0$. Consequently, the network dynamics is simplified to be,

$$\tau \frac{\partial}{\partial t} u_m^n(\theta, t) = -u_m^n(\theta, t) + \sum_{\theta'=-\pi}^{\pi} W_{rc}(\theta, \theta') r_m^n(\theta', t), \ (n = c, o) \tag{A30}$$

$$r_m^n(\theta, t) = \frac{[u_m^n(\theta, t)]_+^2}{1 + \omega D_m^n(t)}, \tag{A31}$$

$$D_m^n(t) = \sum_{\theta'=-\pi}^{\pi} \left( [u_m^n(\theta', t)]_+^2 + J_{int} [u_m^{n'}(\theta', t)]_+^2 \right), \ n' \neq n. \tag{A32}$$

We see that congruent and opposite neurons in the same module compete with each other via divisive normalization, (*Equation A32*), whose effect is to divisively scale down neuronal activities (*Equation A31*). Hence, the divisive normalization only influences the amplitudes of population activities, not the bump shapes. The shapes of population activities $u_m^n(\theta, t)$ and $r_m^n(\theta, t)$ are fully determined by recurrent connections $W_{rc}(\theta, \theta')$. Since the recurrent connection $W_{rc}(\theta, \theta')$ is a von Mises function, and the convolution of two von Mises functions can be approximated by a new von Mises function, we propose the ansatz that neuronal population activities have the von Mises shape in the stationary state, which are written as,

$$u_m^n(\theta | z_m^n(t)) \approx U_m^n \exp\left[ \frac{a}{2} \cos(\theta - z_m^n(t)) - \frac{a}{2} \right], \tag{A33}$$

$$r_m^n(\theta | z_m^n(t)) \approx R_m^n \exp\left[ a \cos(\theta - z_m^n(t)) - a \right], (m = 1, 2; \ n = c, o). \tag{A34}$$

where $U_m^n$ and $R_m^n$ denote, respectively, the heights of synaptic inputs and neuronal firing rates of $n$-type neurons in module $m$. $z_m^n$ denotes the bump location in the feature space.

In order to check the validity of the proposed von Mises ansatz, we substitute *Equations (A33,A34)* into the network dynamics (*Equations A30-A32*), and get the stationary state of the network (see details in the subsequent section), which is

$$U_m^n = \rho J_{rc} \frac{I_0(a)}{e^{a/2} I_0(a/2)} R_m^n, \tag{A35}$$

where $\rho = N/2\pi$ is the neuronal density with $N$ the number of neuron in the group. Meanwhile, substituting the von Mises ansatz into the divisive normalization (*Equations A31,A32*), we get another relationship between $R_m^n$ and $U_m^n$,

$$R_m^n = \frac{U_m^{n\,2}}{1 + 2\pi\omega\rho(U_m^{n\,2} + J_{int} U_m^{n'\,2}) I_0(a) e^{-a}}, (n' \neq n). \tag{A36}$$

Under the condition of no reciprocal and feedforward inputs, there exists a symmetric solution for the heights of congruent and opposite neurons' population responses, that is $U_m^n$ and $R_m^n$. Although an asymmetric solution for the heights of congruent and opposite neurons' responses also exists, we don't consider it in current theoretical study.

Denote the heights of congruent and opposite neurons' responses as $U_m^c = U_m^o \equiv U_m$ and $R_m^c = R_m^o \equiv R_m$, respectively. Combining *Equations (A35, A36)* yields,

$$[2\pi\omega(1+J_{int})\rho I_0(a)I_0(a/2)e^{-a/2}]U^2 - \rho J_{rc}I_0(a)U + e^{a/2}I_0(a/2) = 0, \qquad (A37)$$

whose solution is calculated to be

$$U_m = \frac{\rho J_{rc}I_0(a) \pm \sqrt{(\rho J_{rc}I_0(a))^2 - 8\pi\omega(1+J_{int})\rho I_0(a)I_0(a/2)^2}}{2\pi\omega(1+J_{int})\rho I_0(a)I_0(a/2)e^{-a/2}}. \qquad (A38)$$

$U_m$ has a real value when the recurrent connection strength $J_{rc}$ is larger than a critical value $J_c$, which is given by

$$J_c = \sqrt{\frac{8\pi\omega(1+J_{int})I_0(a/2)^2}{\rho I_0(a)}}. \qquad (A39)$$

This real-value solution of $U_m$ implies that the network holds persistent response without external inputs. $J_c$ is the minimal strength of recurrent connections to hold a persistent activity. Since no persistent activity was observed in multisensory brain areas such as MSTd and VIP, $J_c$ is the upper bound for the recurrent strength $J_{rc}$ in our model.

## Verification of the von Mises ansatz of network activity

Substituting the von Mises ansatz (*Equations A33 and A34*) into the network dynamics (*Equation A30*), we have

$$\text{LHS} = \frac{\tau U_m^n a}{2}\frac{dz_m^n}{dt}\sin(\theta - z_m^n)\exp\left[\frac{a}{2}\cos(\theta - z_m^n) - \frac{a}{2}\right], \qquad (A40)$$

$$\text{RHS} = -U_m^n\exp\left[\frac{a}{2}\cos(\theta - z_m^n) - \frac{a}{2}\right] + \underbrace{\frac{J_{rc}R_m^n e^{-a}}{2\pi I_0(a)}\sum_{\theta'=-\pi}^{\pi}\exp[a\cos(\theta - \theta') + a\cos(\theta' - z_m^n)]d\theta'}_{I_{rc}(\theta)}$$
$$+\alpha\exp\left[\frac{a}{2}\cos(\theta - x) - \frac{a}{2}\right] + \sqrt{\alpha}\exp\left[\frac{a}{4}\cos(\theta - x) - \frac{a}{4}\right]\xi(\theta, t) \qquad (A41)$$
$$+I_{Bkg} + \sqrt{I_{Bkg}}\epsilon(\theta, t).$$

The recurrent inputs $I_{rc}(\theta)$ (the 2nd term in RHS in above equation) can be calculated as,

$$\begin{aligned}
I_{rc}(\theta) &= \frac{J_{rc}R_m^n e^{-a}}{2\pi I_0(a)}\sum_{\theta'=-\pi}^{\pi}\exp[a\cos(\theta - \theta') + a\cos(\theta' - z_m^n)]d\theta' \\
&\approx \frac{\rho J_{rc}R_m^n e^{-a}}{2\pi I_0(a)}\int_{-\pi}^{\pi}\exp[a\cos(\theta - \theta') + a\cos(\theta' - z_m^n)]d\theta' \\
&\approx \rho J_{rc}R_m^n e^{-a}2\pi I_0(a)\mathcal{M}(\theta; z, A^{-1}\{A(a)^2\}) \\
&\approx \rho J_{rc}R_m^n e^{-a}2\pi I_0(a)\mathcal{M}(\theta; z, a/2) \\
&= \rho J_{rc}Re^{-a/2}\frac{I_0(a)}{I_0(a/2)}\exp\left[\frac{a}{2}\cos(\theta - z_m^n) - \frac{a}{2}\right].
\end{aligned} \qquad (A42)$$

The first approximation in the above calculation comes from the conversion from discrete summation to continuous integral, where $\rho = N/2\pi$ is the neuronal density corresponding to the reciprocal of the summation intervals. The last two approximations are from the convolution of two von Mises distributions as given by *Equation (A24)*.

