## [Decision Letter]

Thank you for submitting your article "Complementary congruent and opposite neurons achieve concurrent multisensory integration and segregation" for consideration by *eLife*. Your article has been reviewed by two peer reviewers, one of whom is a member of our Board of Reviewing Editors, and the evaluation has been overseen by Joshua Gold as the Senior Editor. The reviewers have opted to remain anonymous.

The reviewers have discussed the reviews with one another and the Reviewing Editor has drafted this decision to help you prepare a revised submission.

Summary:

It is often the case that the brain receives more than one cue about a quantity of interest. Not surprising, in multisensory areas many neurons have similar tuning to the two cues. Slightly more surprising, some neurons have opposite tuning to the two cues. So far we do not have a clear and unified explanation for the opposite tuning; this paper provides one. The authors extended their previous work (Zhang et al., 2016), and show that the opposite tuned neurons can be used to determine whether or not the two cues are really providing information about the same quantity. They also provided plausible neural circuitry, consisting of a decentralized attractor network, for doing this. This is an important result, and although the material is a bit dense, overall the manuscript is clearly written.

Essential revisions:

We have only one major comment: Despite claims by the authors that their model performs Bayesian inference (e.g.: "Equation 4 is the result of Bayesian optimal integration"), we believe that it doesn't. To perform Bayesian inference, it's necessary to have a prior that allows the two cues to be either the same or different; something like

p(s_1_, s_2_|x_1_, x_2_) ∝ p(x_1_, x_2_|s_1_, s_2_) p(s_1_, s_2_)

= p(x_1_|s_1_) p(x_2_|s_2_) [p_0_δ(s_2_-s_1_) + (1-p_0_) p(s_2_-s_1_)]

where δ(…) is the Dirac δ function. (It's actually a bit more complicated, since it's possible that only one cue is present, and in general the cues should have different amounts of reliability, but an extension to that wouldn't be too hard.) To make contact with the paper, one can integrate over s_2_, yielding

p(s_1_|x_1_, x_2_) = ∫p(s_1_, s_2_|x_1_, x_2_) ds_2_ ∝ p_0_ p(x_1_|s_1_) p(x_2_|s_1_) + (1-p_0_) ∫p(s2-s1)p(x1|s1)p(x2|s2)ds_2_

= p_0_ p(x_1_|s_1_) p(x_2_|s_1_) + (1-p_0_) ∫p(z)p(x1|s1)p(x2|s1+z)dz

If p(z) = δ(z-pi), then

p(s_1_|x_1_, x_2_) ∝ p_0_ p(x_1_|s_1_) p(x_2_|s_1_) + (1-p_0_) p(x_1_|s_1_) p(x_2_|s_1_ + pi).

In this case, one recovers the two terms in the paper: the first term corresponds to Equation 3; the second to Equation 6 (both under a flat prior).

However, it's not the case that p(z) = δ(z-pi); instead, p(z) is, we believe, more or less uniform. In that case the integral over z is probably tractable, although we admit that we haven't checked.

Given the above analysis, we see two possibilities:

1) Admit that this model is reasonable, but it doesn't do Bayesian inference for the cue integration problem.

2) Show that the above analysis, or something like it, does lead (at least approximately) to the network that the authors end up constructing.

Option 2 would be preferable, and we have the feeling that it would be possible, but we would be happy with 1 as well.

Other points:

1) We assume that the preferred direction used in the population vector decoding is the preferred heading of the neuron's major input. We didn't see that stated explicitly (although we may have missed it). It would be worth noting that, for example in the legend in Figure 6 or in Equation 22.

2) Abstract, last sentence: We don't see results that support 'rapid' decision making (compared with what?). Concurrent does not always mean rapid. We would suggest either emphasizing this less, or providing supporting evidence.

3) Introduction paragraph three: We think the concerns about losing information about individual cues during integration is exaggerated. Primary sensory cortices along with working memory may maintain the information, especially if segregation can be done rapidly.

4) Discussion paragraph seven: Neurons responding to center and surround differently may not be good examples. Here, cues are motions in different spatial locations (center vs. surround). In the problem of multisensory integration, different cues encode the same variable (e.g., heading).

5) Discussion paragraph three: Suggested experiments don't seem to test the network structure proposed here. Can you come up with experiments that dissect the network structure? For example, how does activity change in other areas when one area is inactivated? Would you expect to see negative correlations in spiking activity between opposite neurons in the two areas and positive correlations between congruent ones? How about optogenetic inactivation experiments (even if the technique is not fully established in monkeys) that show characteristic rebound activity, as shown in Guo et al., 2017 Nature paper from Svoboda lab?

6) Subsection “Neural encoding model”: In the network, distribution of preferences is uniform. However, the distribution of visual or vestibular preference is bimodal with more neurons preferring lateral headings (Gu et al, 2006). Are the results still consistent in that situation?

7) We're somewhat curious whether there are computational benefits of a decentralized network versus a centralized one. If the authors have some thoughts on this, they would be worth mentioning. But it's not necessary.

---

## [Author Response]

Essential revisions:We have only one major comment: Despite claims by the authors that their model performs Bayesian inference (e.g.: "Equation 4 is the result of Bayesian optimal integration"), we believe that it doesn't. To perform Bayesian inference, it's necessary to have a prior that allows the two cues to be either the same or different; something like

*p(s_1_, s_2_|x_1_, x_2_)* ∝ *p(x_1_, x_2_|s_1_, s_2_) p(s_1_, s_2_)*

= p(x_1_|s_1_) p(x_2_|s_2_) [p_0_δ(s_2_-s_1_) + (1-p_0_) p(s_2_-s_1_)]

where δ(…) is the Dirac δ function. (It's actually a bit more complicated, since it's possible that only one cue is present, and in general the cues should have different amounts of reliability, but an extension to that wouldn't be too hard.) To make contact with the paper, one can integrate over s_2_, yielding

p(s_1_|x_1_, x_2_) = ∫p(s_1_, s_2_|x_1_, x_2_) ds_2_ ∝ p_0_ p(x_1_|s_1_) p(x_2_|s_1_) + (1-p_0_) ∫p(s2-s1)p(x1|s1)p(x2|s2)ds_2_

= p_0_ p(x_1_|s_1_) p(x_2_|s_1_) + (1-p_0_) ∫p(z)p(x1|s1)p(x2|s1+z)dz

If p(z) = δ(z-pi), then

*p(s_1_|x_1_, x_2_)* ∝ *p_0_ p(x_1_|s_1_) p(x_2_|s_1_) + (1-p_0_) p(x_1_|s_1_) p(x_2_|s_1_ + pi).*

In this case, one recovers the two terms in the paper: the first term corresponds to Equation 3; the second to Equation 6 (both under a flat prior).However, it's not the case that p(z) = δ(z-pi); instead, p(z) is, we believe, more or less uniform. In that case the integral over z is probably tractable, although we admit that we haven't checked.Given the above analysis, we see two possibilities:1) Admit that this model is reasonable, but it doesn't do Bayesian inference for the cue integration problem.2) Show that the above analysis, or something like it, does lead (at least approximately) to the network that the authors end up constructing.Option 2 would be preferable, and we have the feeling that it would be possible, but we would be happy with 1 as well.

We appreciate very much the constructive suggestions of the reviewers. We realize that there exists a discrepancy between our interpretation and reviewers’ interpretation on Bayesian integration. In our original manuscript, Bayesian integration is separated from cue segregation, which refers to the optimal integration behavior of the neural system in the congruent cueing condition as observed in the experiments; in the mathematical formulation, this corresponds to estimating the posterior distribution of the stimulus given by the integration prior. Based on the comments, we realize that reviewers are considering a full Bayesian inference for multisensory processing, also termed as causal inference, which includes not only cue integration, but also cue segregation. In this sense, we agree that our model does not achieve Bayesian inference for multisensory processing.

In the present study, we show that opposite neurons can encode the disparity information between cues and also demonstrate that this complementary information can be used to access integration vs. segregation, and that 1) if integration is chosen, the neural system takes the result of congruent neurons which have already integrated multisensory cues based on the integration prior concurrently; 2) if segregation is chosen, the neural system takes the results of individual cues which can be recovered by opposite neurons if necessary. In the present study, we have not explored whether congruent and opposite neurons combine together to achieve the full Bayesian inference for multisensory processing as suggested by the reviewers. There are a number of issues unresolved, including the form of the segregation prior, the network structure for realizing causal inference. We feel that to address these issues, it will distract the reader from the major message of this article and take a lot of work, which we prefer to carry out in the future study. We therefore would like to choose the option 1.

To avoid confusion on the interpretation of Bayesian integration, we have re-phrased words/sentences wherever necessary throughout the paper to emphasize that our model (the congruent neuron part) only implements Bayesian integration using the integration prior, and added a paragraph in Discussion to discuss about the full Bayesian inference for multisensory inference:

“It is worthwhile to point out that in the present study, we have only demonstrated that congruent neurons implement Bayesian cue integration within the framework of a single-component prior and that opposite neurons encode the cue disparity information, and we have not explored whether they can combine together to realize a full Bayesian inference for multisensory processing. In the full Bayesian inference, also termed as the causal inference, the neural system utilizes the prior knowledge about the probabilities of two cues coming from the same or different objects. The prior can be written asp(s1,s2)=∑Cp(s1,s2|C)p(C),where *C* = 1 corresponds to the causal structure of two cues from the same object and *C* = 2 the causal structure of two cues from different objects. The posterior of stimuli is expressed as *p(s*_1_,*s*_2_|*x*_1_,*x*_2_) = Ʃ_c_
*p(s*_1_,*s*_2_|*x*_1_,*x*_2,_
*C) p(C*|*x*_1_,*x*_2_), which requires estimating the causal structure of cues. It is possible that opposite neurons, which encode the cue disparity information, can help the neural system to implement the causal inference. But to fully address this question, we need to resolve a number of issues, including the exact form of the prior, the network structure for realizing model selection, and the relevant experimental evidence, which form our future research.”

Other points:1) We assume that the preferred direction used in the population vector decoding is the preferred heading of the neuron's major input. We didn't see that stated explicitly (although we may have missed it). It would be worth noting that, for example in the legend in Figure 6 or in Equation 22.

Thanks for the suggestion. We actually described this notation in Materials and methods, but as pointed out by the reviewers, this can be easily missed by readers. We therefore add this statement in several other places, including in the legend of Figure 6.

2) Abstract, last sentence: We don't see results that support 'rapid' decision making (compared with what?). Concurrent does not always mean rapid. We would suggest either emphasizing this less, or providing supporting evidence.

Thanks for the suggestion. We agree that the claim of “rapid” is not well justified and delete this claim wherever necessary in the revised manuscript.

3) Introduction paragraph three: We think the concerns about losing information about individual cues during integration is exaggerated. Primary sensory cortices along with working memory may maintain the information, especially if segregation can be done rapidly.

Thanks for the suggestion. We agree that there is a possibility that the brain has other resources to retain the individual cue information. Our study just points out a potential alternative way of using opposite neurons to recover the individual cue information lost in integration. To avoid exaggerating the statement, we describe the opposite neuron solution in a way as hypothesis and discuss about other solutions in Discussion:

“For recovering the stimulus information from direct cues by using the activities of congruent and opposite neurons, this study has shown that it can be done in a biologically plausible neural network, since the operation is expressed as solving the linear equation given by Equation 8. A concern is, however, whether recovering is really needed in practice, since at each module, the neural system may employ an additional group of neurons to retain the stimulus information estimated from the direct cue. An advantage of recovering the lost stimulus information by utilizing congruent and opposite neurons is saving the computational resource, but this needs to be verified by experiments.”

4) Discussion paragraph seven: Neurons responding to center and surround differently may not be good examples. Here, cues are motions in different spatial locations (center vs. surround). In the problem of multisensory integration, different cues encode the same variable (e.g., heading).

Thanks for the suggestion. We agree that the center-surround case is not a good example for comparison. We removed this example and modified the descriptions accordingly:

“Indeed, for example, it has been found in the visual system, there exist ‘what not’ detectors which respond best to discrepancies between cues (analogous to opposite neurons) and they facilitate depth and shape perceptions.”

On the other hand, we would like to point out at here that the underlying math for the center surround example is the same as the multisensory integration. We consider two MT neighbor hyper-columns whose spatial receptive fields are next to each other in the retinotopic map. The inputs received by two hyper-columns can be generated from different parts of the same object moving with the same direction, or different objects moving with different directions. Although the cues presented in the center or surround are at different spatial locations, they can encode the same object. If we describe the whole process by using a generative model, it has the same structure as cue integration. The neural system needs to estimate whether inputs received by two hyper-columns are from the same or different objects, and determines whether to integrate or segregate them.

5) Discussion paragraph three: Suggested experiments don't seem to test the network structure proposed here. Can you come up with experiments that dissect the network structure? For example, how does activity change in other areas when one area is inactivated? Would you expect to see negative correlations in spiking activity between opposite neurons in the two areas and positive correlations between congruent ones? How about optogenetic inactivation experiments (even if the technique is not fully established in monkeys) that show characteristic rebound activity, as shown in Guo et al., 2017 Nature paper from Svoboda lab?

As suggested by reviewers, we have added a small paragraph discussing how to test the key structure of our model in experiments through measuring the correlations between the same and different types of neurons within and across modules, and the effect of inactivating one module to the other:

“The key structure of our network model can be tested in experiments. For instance, we may measure the correlations between congruent neurons and between opposite neurons across modules, and the correlations between congruent and opposite neurons within and across modules. According to the connection structure of our model, the averaged correlations between the same type of neurons across modules are positive due to the excitatory connections between them; whereas the averaged correlations between different types of neurons within and across modules are negative due to the competition between them. We may also inactivate one type of neurons in one module and observe that in the other module, the activity of the same type of neurons is suppressed, whereas the activity of the different type of neurons is enhanced.”

6) Subsection “Neural encoding model”: In the network, distribution of preferences is uniform. However, the distribution of visual or vestibular preference is bimodal with more neurons preferring lateral headings (Gu et al, 2006). Are the results still consistent in that situation?

According to the previous studies (e.g., Ganguli and Simoncelli, Neural Computation 2014; Girshick et al., Nat. Neurosci., 2011), the non-uniform distribution of neurons’ preference is resulted from the non-uniform distribution of heading direction in reality. For simplicity, our work doesn’t consider the non-uniform marginal prior *p*(*s_l_*) for each cue. However, including a non-uniform marginal prior *p*(*s_l_*) will not change our main results. In the below, we briefly derive this conclusion (for clarity, we have not included this detailed explanation in the revised manuscript).

In the probabilistic model, an example of the logarithm of a non-uniform prior of two stimuli can be written as (for simplicity, we consider a Gaussian distribution)ln⁡p(s)=−12(s−μ)TΣp−1(s−μ),wheres=(s1,s2)T,μ=(μ1,μ2)T,Σp−1=(σp1−2+σs−2−σs−2σs−2σp2−2+σs−2)=(σp1−200σp2−2)⏟Λp+σs−2(1−1−11)⏟Λs

This prior can be decomposed into two parts: one is the non-uniform marginal prior of each stimulus, and another is the correlation between two stimuliln⁡p(s)=[ln⁡p(s1)+ln⁡p(s2)]+ln⁡c(s1,s2)=−12(s−μ)TΛp(s−μ)−12(s−μ)TΛs(s−μ)=−(2σp12)−1(s1−μ1)2−(2σp22)−1(s2−μ2)2⏟non−uniformpart−(2σs2)−1(s1−s2)2⏟correlatedpart

The correlated part of the prior c(s1,s2) is similar to the one considered in our study (Eq. 2, consider the analogy between Gaussian and von Mises distribution *N* (μ, σ^2^) ≃ *Μ* (μ, σ^2^)) σs2characterizes the amount of correlation between two stimuli in the prior, which determines the extent of integration; whereas, the variance of marginal prior, σp12and σp22 doesn’t influence the extent of integration.

Note that the non-uniform part of priorln⁡p(s1)+ln⁡p(s2) shares the *similar form* with the likelihood function p(**x**|**s**)ln⁡p(x|s)=ln⁡p(x1|s1)+ln⁡p(x2|s2)=−(2σ12)−1(s1−x1)2−(2σ22)−1(s2−x2)2.

Therefore, the influence of non-uniform marginal prior can be completely absorbed into the likelihood function, i.e.,ln⁡p(xl|sl)+ln⁡p(sl)=−(2σl2σpl2σl2+σpl2)(sl−σpl2xl+σl2μlσl2+σpl2)2.

We see that the effect of the marginal prior is equivalent to changing the mean and reliability of the likelihood. Thus, considering a non-uniform marginal prior won’t change our results on the integration and segregation of multiple cues. In terms of neural computation, the contribution of the non-uniform marginal prior can be implemented through heterogeneous population coding in a module, e.g., the width and density of neurons’ tunings are different across neural populations in MSTd. Our above derivations only consider a single-modal marginal prior. For bimodal prior distribution as pointed out by the reviewers, our model also works as long as the posterior has a strong peak and can be well approximated by a Gaussian distribution.

7) We're somewhat curious whether there are computational benefits of a decentralized network versus a centralized one. If the authors have some thoughts on this, they would be worth mentioning. But it's not necessary.

The cue integration in a decentralized system is collectively emerged from the interactions among modules. One advantage of the decentralized system over the centralized one is the robustness to local failures. Experimental data has indicated that the monkey can still perform optimal integration even if inactivating one visual-vestibular area (Gu et al., J. Neurosci., 2012), supporting the decentralized structure.

We added a few sentences describing the advantages of the decentralized system in the revised manuscript.